# Cell-DINO: Self-supervised image-based embeddings for cell fluorescent microscopy

Théo Moutakanni[1,2], Camille Couprie[1], Seungeun Yi[1], Michael Doron[3], Zitong S. Chen[3], Nikita Moshkov[4], Elouan Gardes[1], Mathilde Caron[5], Hugo Touvron[1], Armand Joulin[5], Piotr Bojanowski[1], Wolfgang M. Pernice[6], Juan C. Caicedo[7,8]*

1 Meta Platforms Inc., FAIR, Paris, France, 2 MICS, CentraleSupelec, Université Paris-Saclay, Gif-sur-Yvette, France, 3 Broad Institute, Cambridge, Massachusetts, United States of America, 4 Helmholtz Munich, Institute of Computational Biology, Neuherberg, Germany, 5 Google LLC DeepMind, Paris, France, 6 Columbia University Irving Medical Center, Neurology, New York, New York, United States of America, 7 Morgridge Institute for Research, Madison, Wisconsin, United States of America, 8 Department of Biostatistics and Medical Informatics, University of Wisconsin–Madison, Madison, Wisconsin, United States of America

* juan.caicedo@wisc.edu

## Abstract

Accurately quantifying cellular morphology at scale could substantially empower existing single-cell approaches. However, measuring cell morphology remains an active field of research, which has inspired multiple computer vision algorithms over the years. Here, we show that DINOv2, a vision-transformer based, self-supervised algorithm, has a remarkable ability for learning rich representations of cellular morphology without manual annotations or any other type of supervision. We apply DINOv2 to cell phenotyping problems, and compare the performance of resulting models, called Cell-DINO models, on a wide variety of tasks across two publicly available imaging datasets of diverse specifications and biological focus. Compared to supervised and other self-supervised baselines, Cell-DINO models demonstrate improved performance, especially in low annotation regimes. For instance, to classify protein localization using only 1% of annotations on a challenging single-cell dataset, Cell-DINO performs 70% better than a supervised strategy, and 24% better than another self-supervised alternative. The results show that Cell-DINO can support the study of unknown biological variation, including single-cell heterogeneity and relationships between experimental conditions, making it an excellent tool for image-based biological discovery.

## Author summary

Cellular imaging is widely used in biological studies, and often enables quantitative analysis of phenotypic variation through the use of machine learning algorithms.

**Data availability statement:** All the data used in this work has been made publicly available by their creators. The Human Protein Atlas (https://www.proteinatlas.org/humanproteome/subcellular) and the Cell Painting Gallery (https://broadinstitute.github.io/cellpainting-gallery/overview.html) have been created and are maintained by other institutions. The links above provide more information on how to obtain the original datasets. We downloaded and pre-processed these image datasets, and we provide the transformed version used in our study in BioImage Archive, as follows: Accession: S-BIAD2443; DOI: 10.6019/S-BIAD2443; URL: https://www.ebi.ac.uk/biostudies/bioimages/studies/S-BIAD2443. Our implementation is an adaptation of DINOv2, which is hosted in the main DINOv2 repository. This implementation also includes code to reproduce the results of the HPA analysis. The code is available in GitHub at https://github.com/facebookresearch/dinov2/blob/main/docs/README_CELL_DINO.md. The code necessary to reproduce the Cell Painting results with the LINCS dataset is publicly available in GitHub at: https://github.com/CaicedoLab/cell-painting-devbench.

**Funding:** This work was partially supported by Muscular Dystrophy Association (Development Grant 628114 to WMP) and by the National Institutes of Health (awards K99HG011488 and R00HG011488 to WMP). NM was funded by PD23 OTKA (no. 146881) by the Hungarian Ministry of Innovation. This research was funded by the U.S. National Science Foundation's Directorate for Biological Sciences under Grant No. 2348683 awarded to JCC. The funders had no role in study design, data collection and analysis, decision to publish, or preparation of the manuscript.

In this work, we introduce Cell-DINO, an adaptation of the DINOv2, self-supervised representation learning strategy, and demonstrate its ability to produce better features for cell phenotyping problems. On the Human Protein Atlas (HPA) dataset, our experiments show the versatility of the approach, outperforming a supervised vision transformer strategy by 20% on average and by 70% when using only 1% of the labels. Compared to other self-supervised strategies, Cell-DINO outperforms MAE and SimCLR on the most challenging tasks, for instance protein localization classification on the HPA single-cell dataset, with 34% and 20% improvement in performance, respectively. Cell-DINO can also be useful to quantify the effects of pharmacological compounds at high-throughput, where the phenotypes of interest cannot be annotated by hand. Cell-DINO improves the ability to predict the mechanism of action of compounds using Cell Painting images, outperforming SimCLR, MAE, and supervised CNN baselines, as well as CellProfiler.

## Introduction

Visual interpretation of cellular phenotypes through microscopy is widely used to drive biomedical discovery across a variety of questions in biology, including the assessment of subcellular protein localization [1,2], mitochondrial phenotypes [3,4], and cell-cycle stages [5,6], as well as chemical [7] and genetic perturbations [8,9]. Overcoming the biases and limitations of conventional image analysis methods [10–12], machine learning has the potential to yield unprecedentedly rich and comprehensive features of image-based cellular phenotypes, and to establish morphological profiling among the current roster of single-cell omics in its own right [13–16]. Indeed, deep learning now powers robust cell segmentation methods [17–23], as well as cell phenotyping using classification networks [24,25].

Approaches to image analysis in microscopy remained specialized for a long time. Image analysis in biology has historically relied on bespoke workflows to measure one or two features [26], such as cell counts or cell size, disregarding much of the content present in microscopy images. Over the last decade, image-based profiling appeared with the goal of automating the extraction of large sets of features to quantify morphology more broadly [27,28]. Although widely adopted, this approach still relied on classical features engineered depending on the study. Recently, deep learning has been successful in learning morphological features directly from data, yet these efforts typically relied on extensive manual annotations [29,30] and/or prior knowledge about the biology of interest [31–33], which may limit its use to targeted applications only. By contrast, self-supervised learning (SSL) for vision has recently emerged as a strategy for obtaining rich representations of cellular phenotypes from images [34–37], and to make image-based phenotypes compatible with large-scale perturbation studies where prior knowledge of phenotypic variation may not exist because the goal is to uncover the effect of such perturbations [38–40]. While early

results are promising, the comparative performance and utility of various SSL methods has thus far only been explored on single datasets and/or limited sets of tasks. Moreover, despite their potential, SSL methods typically lag behind the task-specific performance of supervised models.

In our previous work, we designed the DINO algorithm [35] to create foundation models for natural RGB images. Our approach has been shown to discover meaningful factors of variation in image collections, and to achieve state-of-the-art performance in multiple vision problems [41–44]. Innovations introduced with DINOv2 [37] include the use of new loss functions [45,46] to further bolster performance. Here, we leverage these advances in self-supervision to obtain image-based embeddings that measure as many aspects of cellular morphology as possible, without targeting specific phenotypic properties ahead of time, by applying the DINOv2 training algorithm [37] to images of cells. We refer to the approach and resulting models as *Cell-DINO* (Fig 1). Our study shows that Cell-DINO is an effective self-supervised approach capable of uncovering meaningful biology from fluorescent microscopy images without the need for manual annotations or explicit supervision.

In this work, we propose that Cell-DINO can be used as a general-purpose strategy for transforming images of cells into quantitative phenotypic data (image embeddings) to advance experimental biology with the use of microscopy. We analyzed two publicly available datasets with different technical specifications and biological goals (Fig 2), and obtained results that show how the embeddings produced by Cell-DINO support a wide range of downstream tasks, from subcellular to population-level analyses. The datasets and tasks evaluated in our study are representative of phenotyping and image analysis problems that permeate biomedical research practice, and for which extensive prior work exists, including strong classical feature analysis strategies, and specialized deep learning solutions. Out of the scope of this work is the training of foundation models that generalize beyond the reported benchmarks; instead, our focus is to investigate the properties of Cell-DINO and purely self-supervised strategies to extract meaningful information from biological images. We compare Cell-DINO against relevant state-of-the-art solutions, as well as contemporary SSL strategies. Our results indicate that DINOv2 has great potential for powering various specific bioimage analysis applications [39,41,88] and to enable image-based biological discovery.

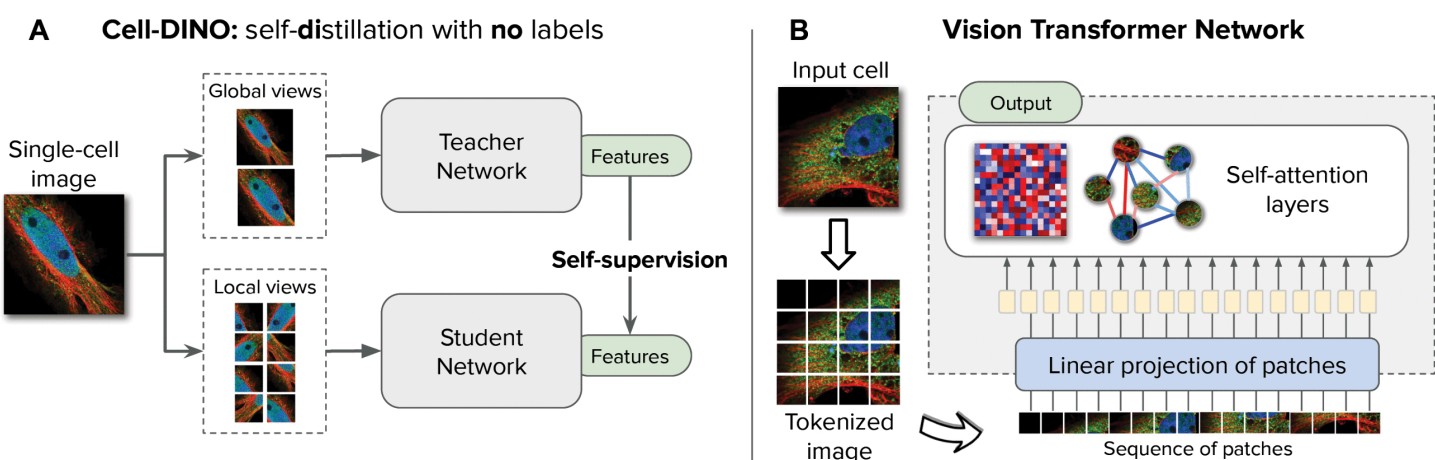

**Fig 1**. **Cell-DINO: self-distillation with no labels, vision transformers, and datasets.** A) Illustration of the Cell-DINO algorithm, which trains two neural networks: a teacher and a student network. The self-supervised objective aims to match the features produced by the teacher (which observes random global views) with those produced by the student (which observes random global and local views). B) Illustration of the processing pipeline using vision transformer networks trained with DINO. The image of a cell is first tokenized as a sequence of local patches, which the transformer processes using self-attention layers. The output are image embeddings used for downstream tasks.

## Results

We investigate the capabilities of Cell-DINO in two separate image-based cell phenotyping problems: 1) protein localization and 2) compound screening. The goal of the protein localization task is to identify the subcellular compartment in which a protein of interest is observed in images of cells. For this problem, we use the Human Protein Atlas (HPA) dataset [1], which has been used to benchmark machine learning algorithms in two Kaggle competitions [24,25] and also has extensive manual annotations. The goal of the compound screening task on the other hand is to quantify similarities and differences in the response of cells to treatments by detecting relevant morphological changes. For this problem, we used publicly available datasets from the Cell Painting Gallery (CPG) [47] acquired to screen chemical and genetic perturbations in cancer cell lines. The CPG datasets have fewer annotations and are representative of contemporary image-based profiling experiments.

### Cell-DINO encodes diverse biological properties of cells

We used our Cell-DINO approach to analyze the HPA and CPG datasets (Fig 2), and we observe that it discovers meaningful factors of variation directly from the images in both cases. In particular, we trained individual Vision Transformers (ViT) using the DINOv2 algorithm on HPA field-of-view (FoV) images (containing multiple cells), as well as single-cell (SC) images for both HPA and CPG, after deriving single-cell images via segmentation algorithms optimized for each case (Methods). We find that the resulting Cell-DINO models yield image embeddings that capture the phenotypic diversity of cells in the corresponding dataset. Specifically, we find that Cell-DINO identifies clusters of cell lines in the HPA-FoV dataset (Fig 3A), and cell lines and perturbation type in the CPG datasets as primary factors of variation (Fig 3B). These results illustrate how Cell-DINO learns meaningful biological phenotypes without being explicitly trained to identify or detect them.

In contrast to supervised models that specialize on specific tasks, the self-supervised learning algorithm DINOv2 may discover phenotypic structure beyond these targeted factors of variation. That could enable the analysis of multiple phenotypic questions based on a single set of rich, task-agnostic features. To explore this, we transformed the Cell-DINO HPA-FoV embeddings using the single-cell data integration algorithm Harmony [52] to remove variation across cell-line clusters. We point out that Harmony receives only cell-line labels, but no information on protein localization labels. Remarkably, visualizing the resulting harmonized embeddings uncovers a coherent substructure of protein localization clusters,

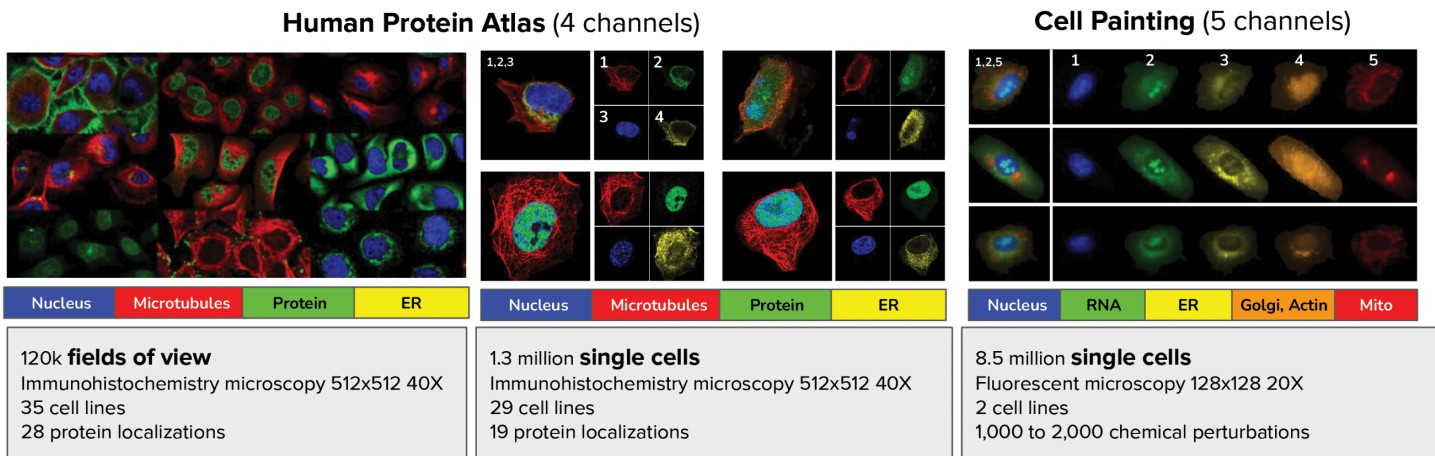

**Fig 2. Datasets used in this study.** Example images of the datasets used in our evaluation: the Human Protein Atlas Field-of-View (FoV) and Single-Cell (SC) datasets, and a collection of Cell Painting datasets.

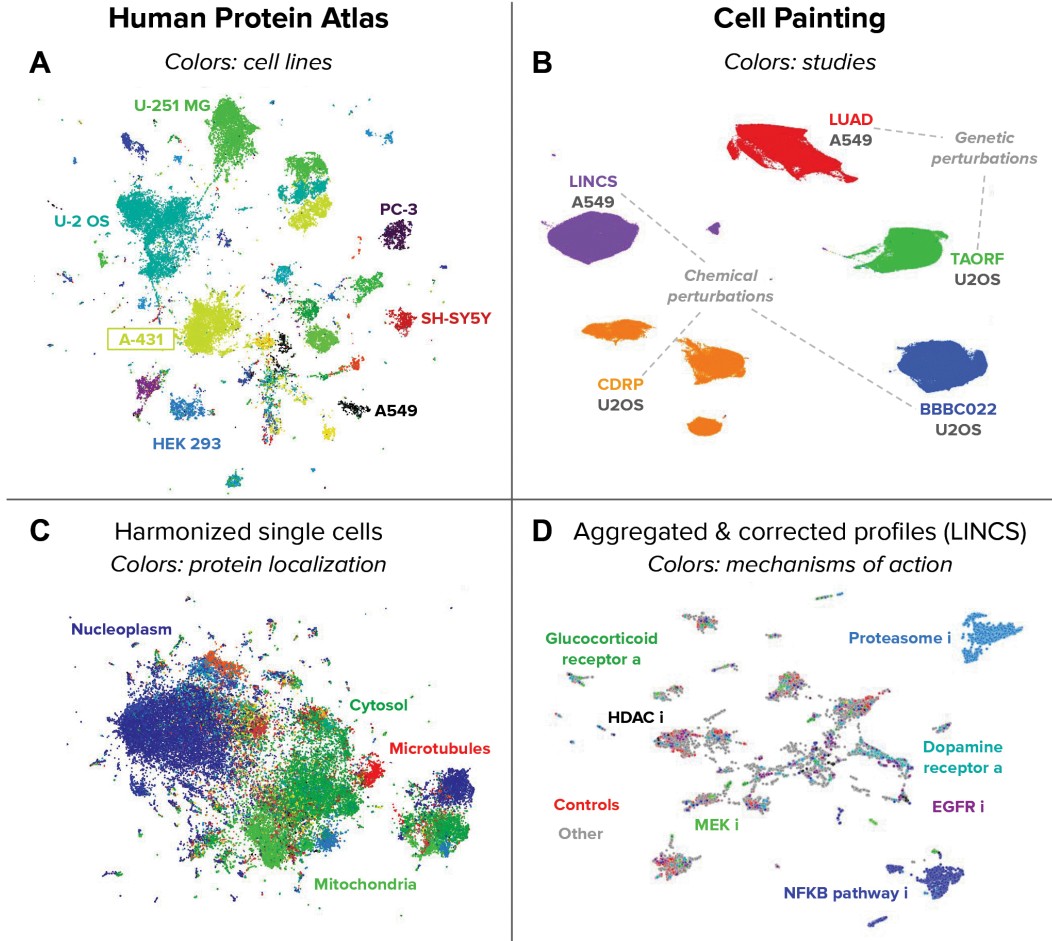

**Fig 3**. **UMAP visualizations of image-based embeddings obtained with Cell-DINO** Left column (A,C): Image-based embeddings of the HPA-FoV dataset. Right column (B,D): Image-based embeddings of the Cell Painting datasets. Top row (A,B): unprocessed embeddings. Bottom row (C,D): transformed embeddings for downstream analysis. A) Points are fields of view, and the colors reveal the cell line of the sample. B) Points are single cells, and the colors reveal the source study where the cells come from: LINCS [48] and LUAD [9] are A549 cells, while CDRP [49], TAORF [50], and BBBC022 [51] are U2OS cells. Samples from the five studies were used for training. C) Points are embeddings of fields of view after harmonization [52]; colors are protein localization labels. D) Points are well-level aggregated features after batch correction with sphering [73] from the LINCS dataset. Colors are mechanisms of action; names with **a** are antagonists and names with **i** are inhibitors).

supporting the idea that protein localization patterns may be consistent across human cell lines [53] (Fig 3C). This in turn demonstrates Cell-DINO's ability to encode biologically meaningful properties of cells in a way that can be flexibly disentangled for downstream analysis. Note that the resulting harmonized HPA-FoV embeddings qualitatively resemble the structure identified by supervised methods that are trained specifically for protein localization while ignoring cell lines [24] (Fig A in S1 Text). We next asked how Cell-DINO's embeddings compare to specialized baselines on quantitative tasks.

## Cell-DINO outperforms standard supervised ViT

Vision transformers are considered today as the leading architecture in computer vision, reaching state-of-the-art performance on a number of tasks, including image classification [54], object detection [55], and semantic segmentation [56], among others. Cell-DINO is trained with self-supervision, requiring no manual annotations to learn a ViT model that computes meaningful image embeddings. Not using labels for training can be considered a disadvantage because the model

is not informed about the relevant image patterns for downstream analysis. Contrary to this, we find that Cell-DINO models can outperform standard supervised ViT models by a substantial margin. Our supervised implementation follows the original classification strategy for ViTs [57], leaving out other potential tricks or advanced supervised optimizations that may improve performance. Specifically, we benchmark Cell-DINO's performance on several phenotyping tasks in the HPA datasets, which cover a diversity of subcellular protein localization categories observed in multiple cell lines. Note that the results described in this section (Fig 4) were obtained using a routine train/validation split of the HPA datasets, a more canonical use case for most practitioners, whereas the Kaggle competition test images present a specific generalization task [24], which we pursue in separate evaluations in the following section.

Prior work on HPA [24,25] trained supervised deep learning methods directly on protein localization labels, which forces the models to ignore other aspects of cellular morphology –such as broader phenotypes that characterize cell lines– to maximize performance. The intuition is that cell-line variation may be a distractor that prevents the correct identification

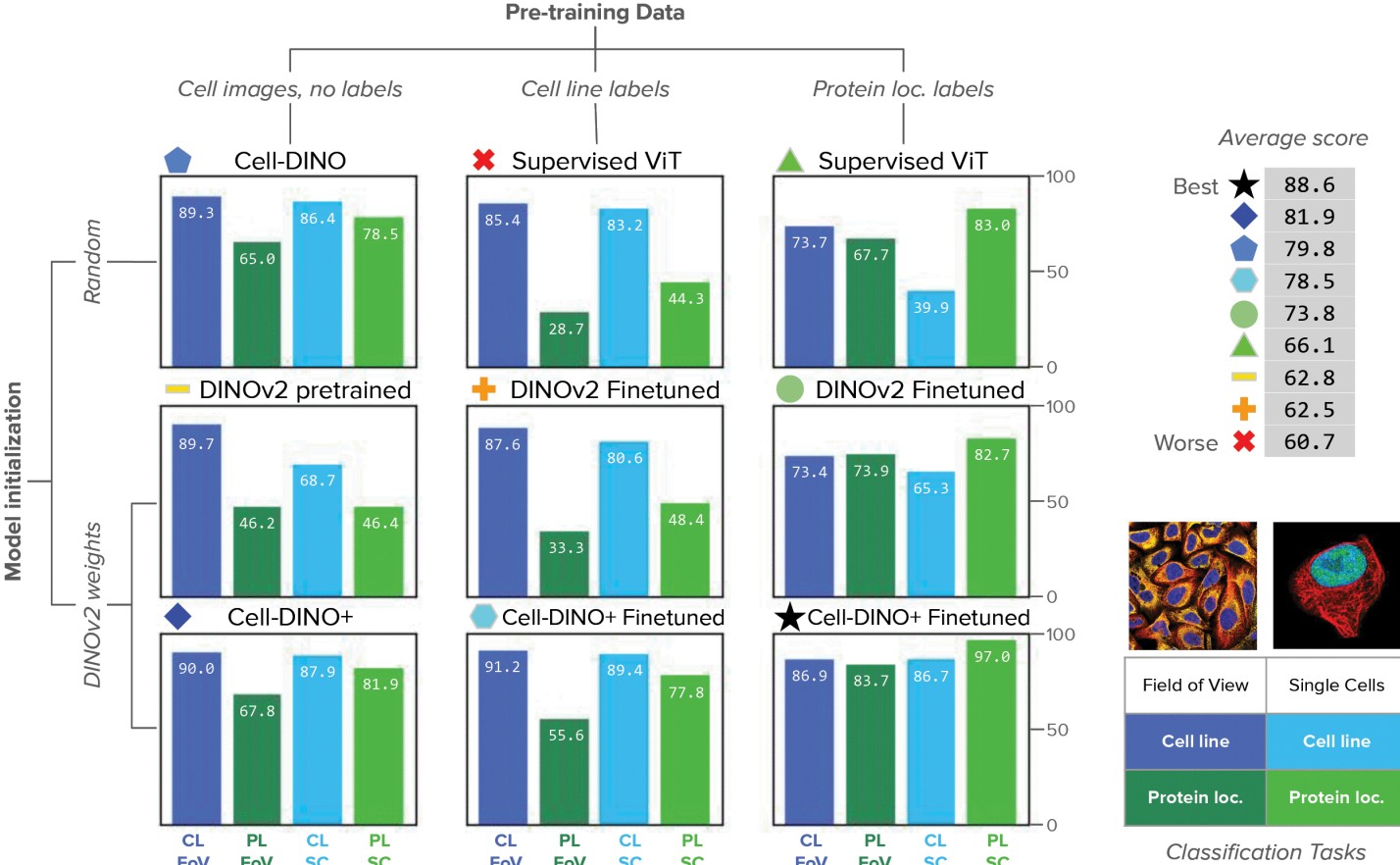

**Fig 4. Performance evaluation of Cell-DINO on the HPA dataset.** All reported results are the F1-scores of the corresponding classification task. There are two classification tasks: protein localization (PL/green shades) and cell-line (CL/blue shades) prediction. Each task is evaluated in two versions of the HPA dataset: field-of-view (dark colors) images and single-cell images (light colors). See legend in the bottom-right. Results are presented in a grid of bar-plots colored according to the task and dataset, and organized by pretraining data (columns) and model initialization (rows). Pre-training data goes from no supervision with any labels (left column), to medium supervision with cell line labels (center column), and strong supervision with protein localization labels (right column). The rows consider random initialization (top row), pre-trained DINOv2 models without self-supervision with cell images (center row), and initialization with DINOv2 pre-trained weights with additional self-supervision with cellular images (bottom row). The average score in the top-right corner of the figure identifies each method with a colored icon according to the ranking from best to worst performance. The results of Cell-DINO fine-tuned are reported in Table C in S1 Text.

of the phenotype of interest (protein localization), therefore, an optimal model will encode only those features. To test the extent to which Cell-DINO's embeddings can compete with supervised models, we approach cell-line classification and protein localization as two separate tasks. To evaluate these tasks, we compare a single Cell-DINO model against task-specialized supervised ViT models.

We find that classifiers trained with Cell-DINO embeddings obtain excellent performance in both cell-line and protein-localization classification tasks compared to fully supervised ViT models (Fig 4, first row). This shows that Cell-DINO learns substantially richer embeddings than supervised models, rendering it useful for multiple downstream tasks, whereas supervised models fail to generalize well across tasks. This limitation of supervised models is further compounded by the amount of labeled data required to train such models. Second, Cell-DINO embeddings also exhibit performance superior to DINOv2 models pre-trained using natural images and fine-tuned for cell-line or protein-localization classification respectively (Fig 4, second row). This shows that the self-supervised strategy of Cell-DINO can leverage a cellular imaging dataset to discover more biologically relevant features than what pre-trained, state-of-the-art natural image models can extract.

Notably, we find that by initializing the weights of a Cell-DINO model with DINOv2 pre-trained weights (Cell-DINO +), the overall accuracy can be further improved (Fig 4, third row). This result points to an easy way of improving self-supervised results, and we highly recommend its adoption in practice. In the rest of this paper, we compare other baselines and strategies against Cell-DINO instead of Cell-DINO + because the latter has the unfair advantage of being pre-trained with more data (150M+ natural images) to which other methods do not have access.

## Cell-DINO is competitive against highly tuned models

We next ask whether Cell-DINO models can compete with highly optimized models on specific tasks. In particular, both the HPA-FoV and HPA-SC datasets were originally introduced as part of Kaggle competitions with the objective of automating the classification of protein localization [24]. After hundreds of submissions, the leaderboards are dominated by highly-tuned ensembles of supervised models, leveraging complex losses, custom class weighting, and data curation, rendering this benchmark very challenging for self-supervised approaches. To test the utility of Cell-DINO's embeddings in this scenario, we trained a small supervised two-layer classifier on the protein localization task using frozen Cell-DINO embeddings as input. Remarkably, we find that Cell-DINO ranks in the top 8.8% and 11.6% of submissions of FoV and SC competitions, respectively, among hundreds of highly specialized submissions (Fig 5), even though Cell-DINO does not have specific inductive biases to solve this problem. We further compared Cell-DINO embeddings, against embeddings derived from DINOv2, and the 1st and 2nd ranked solutions in the SC and FoV classification challenges, respectively (Bestfitting [24] and DualHead [25]). Our results confirm that Cell-DINO learns features that perform comparably with fully supervised models in solving protein localization classification, exhibiting negligible performance differences (only 0.014 and 0.077 absolute points in the FoV and SC challenges respectively, Fig 5).

## Cell-DINO outperforms alternative self-supervised strategies

We compared Cell-DINO to other two main families of SSL methods, including reconstructive autoencoders, and contrastive learning. The Masked Autoencoder (MAE) [36] approach uses an image reconstruction objective and has been previously shown to obtain scalable results on cellular imaging tasks [38]. The SimCLR approach [34] introduces a contrastive loss maximizing the agreement between two different transformations of the same images in the feature space. MAE and SimCLR are considered strong self-supervised approaches on standard computer vision benchmarks; MAE is widely used by the community, and often requires fine-tuning or additional supervision, while SimCLR has been shown to scale well with more data and larger models [58]. Other methods such as MoCo [59] (contrastive), BYOL [60] (self-distillation), and SimSiam [61] (contrastive) fall into one of the main SSL families evaluated in our work, and have been

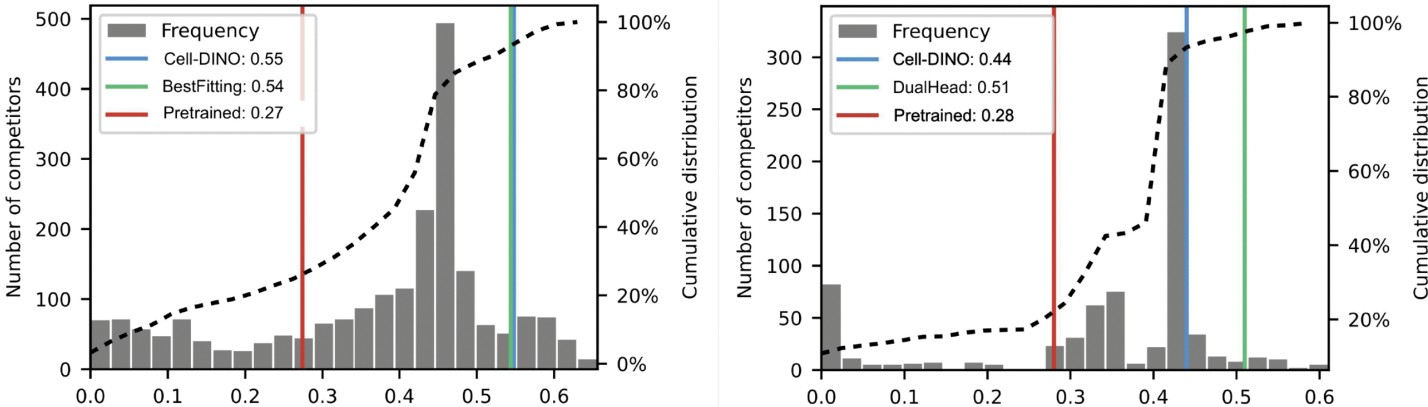

**Fig 5**. **Evaluation of Cell-DINO in the HPA Kaggle competitions.** Left: whole image protein classification competition [24]. Right: weakly-supervised, single-cell protein classification competition [25]. Both plots: distribution of performance scores in the second stage evaluation of valid competition submissions. The horizontal axis is the official competition score (F1-score on the test set), and the vertical axis is the frequency of submissions obtaining the corresponding score. The dotted line is the cumulative distribution. Vertical lines represent models evaluated in this work as follows: ImageNet pre-trained DINO (red), a single supervised CNN from the top competitors (green), Cell-DINO (blue).

found to perform on par to each other (BYOL, MOCOv2, SimSiam [61]), or to be outperformed by DINO [35] (MoCov2 and BYOL) on natural image benchmarks.

As reported in Table 1, we note that the more challenging the task is, the stronger Cell-DINO outperforms these models. In particular, on the most challenging HPA-SC dataset, Cell-DINO achieves a boost of more than 6 points in cell-line classification and 13 points in protein localization. By contrast, without finetuning, MAE achieves poor results on the protein localization task, underperforming Cell-DINO by almost 50%.

We also conducted a quantitative evaluation of the predictive performance of Cell-DINO in the CPG datasets by approaching two different tasks. The first task is mechanism-of-action (MoA) prediction in chemical perturbation experiments, following a transcriptomics-based prediction strategy previously used in a Kaggle competition and adapted for Cell Painting images [62]. The second task is treatment prediction, which serves as an auxiliary task for representation learning [63], and has been adapted in previous work for evaluation in the JUMP Cell Painting dataset [15]. Our results show that Cell-DINO robustly outperforms both SimCLR and MAE SSL algorithms in both tasks (Table 1), and is even able to improve upon the performance of previous state-of-the-art approaches, including classical features and weakly supervised learning [32]. These results confirm that Cell-DINO yields substantially richer representations than alternative approaches, yielding a single set of embeddings that improve performance across multiple tasks.

## Cell-DINO outperforms other pre-trained models

There are growing efforts to pre-train and share models for cellular image analysis, including cell segmentation models [19,22] and models for feature extraction [32,33,38,83]. Here we compare the performance of Cell-DINO against state-of-the-art pre-trained models for image-based profiling, specifically OpenPhenom [38] and SubCell [33]. These two are representative of recent developments in the field and bring meaningful innovations to bioimage analysis. Specifically, OpenPhenom is a channel-adaptive ViT pre-trained with Cell Painting data using an extension of the MAE algorithm. Also, SubCell is a family of models pre-trained using MAE and multi-task supervised objectives using the HPA dataset.

**Table 1. Comparison to other self-supervised algorithms.** Top and middle: F1-scores obtained in the HPA-FoV and HPA-SC datasets, respectively, on the protein localization (PL) classification and cell line (CL) prediction tasks. Bottom: Results using Cell Painting images on the LINCS mechanism-of-action (MoA) prediction task using the Area Under the Precision-Recall Curve (AUPRC) metric [62], and on treatment classification using a subset of the JUMP-CP dataset (accuracy) [63].

| HPA-FoV | Pretrain data | Architecture | PL | CL |
|---|---|---|---|---|
| MAE | HPA-FoV | ViT-L | 38.6 | **90.6** |
| SimCLR | HPA-FoV | ViT-L | 62.9 | **90.6** |
| Cell-DINO | HPA-FoV | ViT-L | **65.5** | 89.3 |
| **HPA-SC** | **Pretrain data** | **Architecture** | **PL** | **CL** |
| MAE | HPA-SC | ViT-L | 58.2 | 72.7 |
| SimCLR | HPA-SC | ViT-L | 65.2 | 79.0 |
| Cell-DINO | HPA-SC | ViT-L | **78.5** | **86.4** |
| **Cell Painting** | **Pretrain data** | **Architecture** | **MoA AUPRC** | **JUMP-CP** |
| MAE | Comb Cell Painting | ViT-S | 6.53 | 23.7 |
| SimCLR | Comb Cell Painting | ViT-S | 6.82 | 25.4 |
| Cell-DINO | Comb Cell Painting | ViT-S | **7.67** | **29.4** |

For this evaluation, we use the models released by the authors of these studies to extract features in the benchmark datasets of our study. Table 2 reports the results and the associated details. Note that these models have been pre-trained with different datasets and have different architectures from our models. We adopted their pre-trained models as-is because they have been presented as foundation models that can be used out-of-the-box. In contrast, our models are specific to the datasets used for analysis since our goal is to investigate the capabilities of the Cell-DINO algorithm rather than producing generalizable foundation models. The results indicate that our dataset-specific Cell-DINO models yield better performance in all cases, demonstrating the remarkable ability of our strategy to identify meaningful features from a dataset without supervision. We used OpenPhenom for HPA-FoV and not for HPA-SC because it was trained with multi-cellular image crops. Similarly, we used SubCell for HPA-SC and not for HPA-FoV because the model was trained with single-cells. We used both for the LINCS benchmark because the original studies report adaptations and recommendations for Cell Painting, which we followed in our experiments.

In this evaluation, the baseline models may have a disadvantage because of the difference in training datasets; additional results to address these differences in the HPA dataset are reported in the Tables A and B in S1 Text. The results are nevertheless informative to assess the achievements of the Cell-DINO strategy with respect to the state-of-the-art in the field. The results also open opportunities to combine innovations from these models with the robustness of Cell-DINO in future work.

**Table 2. Comparison to other state-of-the-art pre-trained models.** Top and middle: F1-scores obtained in the HPA-FoV and HPA-SC datasets, respectively, on the protein localization (PL) classification and cell line (CL) prediction tasks. Bottom: Results using Cell Painting images on the LINCS mechanism-of-action prediction task (MoA AUPRC) [62].

| HPA-FoV | Pretrain data | Architecture | PL | CL |
|---|---|---|---|---|
| OpenPhenom [38] | RxRx3, JUMP-CP | ViT-S | 23.2 | 81.7 |
| Cell-DINO | HPA-FoV | ViT-L | **65.5** | **89.3** |
| **HPA-SC** | **Pretrain data** | **Architecture** | **PL** | **CL** |
| SubCell [33] | HPAv23 | ViT-B | 63.6 | 73.4 |
| Cell-DINO | HPA-SC | ViT-L | **78.5** | **86.4** |
| **Cell Painting** | **Pretrain data** | **Architecture** | **MoA AUPRC** | |
| OpenPhenom [38] | RxRx3, JUMP-CP | ViT-S | 6.40 | |
| SubCell [33] | HPAv23 | ViT-B | 7.65 | |
| Cell-DINO | Comb Cell Painting | ViT-S | **7.67** | |

## Cell-DINO reduces the dependency on manual annotations

We further find that Cell-DINO drastically reduces the need for supervised training. Our competitive results in the HPA-SC Kaggle competition test sets (see Fig 5) were obtained using a 2-layer classifier, containing over seven times fewer supervised parameters than even a *single* network inside the ensemble of Bestfitting (1.2 vs. $9.1 \times 10^6$ parameters) [24,25]. We asked whether Cell-DINO may alleviate the need for manual annotations. Importantly, while early progress in the adoption of deep learning methods for cellular imaging applications has been driven by manual annotation efforts, our ability to generate imaging data vastly outpaces biologists' ability to manually annotate phenotypes of interest reliably. To realize the full potential of morphological profiling as a high-content omic, we need methods that can leverage imaging data at scale, while requiring little to no supervised signal.

As such, we assess the robustness of Cell-DINO and alternative approaches to diminishing numbers of labeled images during training by comparing classification performance as measured by the F1-score (Fig 6). We use a supervised baseline, consisting of a ViT encoder backbone and a linear decoder that are trained end-to-end over reduced sets of labeled images. For SSL methods we train the ViT encoders with all available images – even unlabeled – followed by training a linear layer with the same reduced set of labeled images as the supervised baseline. The results indicate that Cell-DINO affords major advantages when the availability of supervised signal is drastically reduced. SimCLR performs on par or slightly better than Cell-DINO in the FoV classification task. However, on the more challenging HPA-SC dataset, Cell-DINO outperforms both supervised and SSL approaches by a substantial margin with only 1% of the available annotations.

Note that at the FoV level, a protein localization class is considered valid if at least one cell in the FoV has the localization of interest. It is usually the case that multiple cells exhibit the localization, resulting in more evidence to select the correct class. Protein localization at the single-cell level is generally more challenging because the pattern needs to be recognized in one cell or the correct classification will be missed. In addition, single-cell training images are not fully annotated, but rather have a weak label inherited from the FoV. Thus, a cell may have a weak positive label assigned when it is actually a negative example. This inconsistency makes it harder to learn from single-cell annotations, and this was by design in the original single cell Kaggle competition because of the difficulty and the challenges of annotating single cells manually [25].

In all cases at 100% of labeled sample usage, the supervised model has access to all the annotated data to tune all of its parameters. In contrast, Cell-DINO only trains the classification head with supervision and the rest of the backbone parameters are pre-trained with self-supervision and then frozen (no fine-tuning). This large difference in supervised parameters would give an advantage to the supervised approach. However, the Cell-DINO models meet and sometimes surpass the performance of supervised models, highlighting the benefits of self-supervision. These include data efficiency that maximizes learning without labels first, and prevention of potential overfitting issues that facilitates generalization.

## Cell-DINO enables predictions in image-based perturbation experiments

Imaging is being extensively used to quantify the effects of pharmacological compounds in high-throughput screens, in which phenotypes of interest cannot be annotated by hand [15]. The CPG datasets are an example of such an application. Here, existing knowledge from chemical biology is used to create predictive models of various aspects of compound bioactivity by looking at the response of cells to perturbations in images [65,66]. We find that Cell-DINO improves the ability of such models to predict one such aspect of particular importance: the mechanism of action (MoA) of compounds based on Cell Painting images of cells treated with them [67]. Specifically, we first trained a Cell-DINO model using a combined Cell Painting dataset, previously curated for representation learning [32]. Next, we computed image embeddings and created well-level representations for the LINCS dataset to predict MoAs [62].

For the prediction of MoA, we leveraged compound annotations from the Connectivity Map project, which were previously used in a Kaggle competition to predict compound MoA based on transcriptional profiles of cells [68]. We adapted

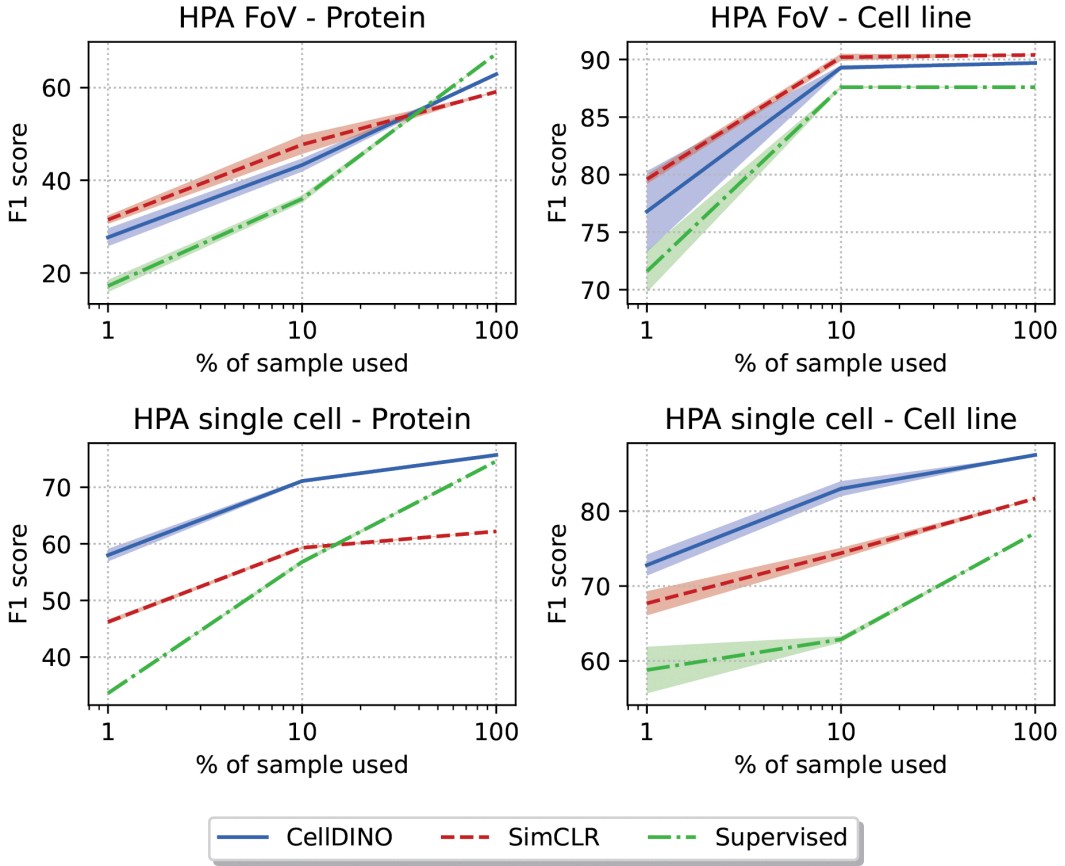

**Fig 6. Low shot performance on HPA datasets.** Top row: results on the HPA-FoV dataset. Bottom row: results on the HPA-SC dataset. Left column: results of the protein localization classification task. Right column: results of the cell-line prediction task. All plots: the horizontal axis is the percent of labeled images used for training in logarithmic scale. The vertical axis is the F1-score of the corresponding classification task.

the same evaluation protocol for image-based profiling based on the LINCS Cell Painting dataset [62], a library of more than 1,500 FDA approved drugs screened on A549 cells. We then used the same profiling pipeline and classifier models (Methods) to compare three feature representations: engineered features obtained with CellProfiler [69,70], features obtained with the weakly supervised Cell Painting CNN model [32,71], and Cell-DINO features trained on Cell Painting images. The results show that learned embeddings improve MoA classification compared to engineered features, and Cell-DINO's embeddings perform better than the CP-CNN model (Fig 7). MoA classification is approached at the well-level; single-cell heterogeneity could potentially improve prediction performance even further. The results are based on the maximum dose; analysis of lower doses are reported in Table D in S1 Text.

## Cell-DINO excels at image-based profiling of cellular morphology

Image-based profiling of biological conditions aims to uncover the relationships among groups of samples to discover their similarities or differences. Cell-DINO features appear useful for image-based profiling, as indicated by their ability to cluster samples in a biologically meaningful manner (Fig 3). We thus evaluate the ability of Cell-DINO embeddings to correctly match samples that are expected to have the same biological annotations using simple nearest-neighbors search. In the HPA datasets, we approach the protein localization and cell line classification tasks using a nearest neighbor classifier (kNN), in contrast to previous work that uses full CNN models or linear classifiers. This is a significantly more

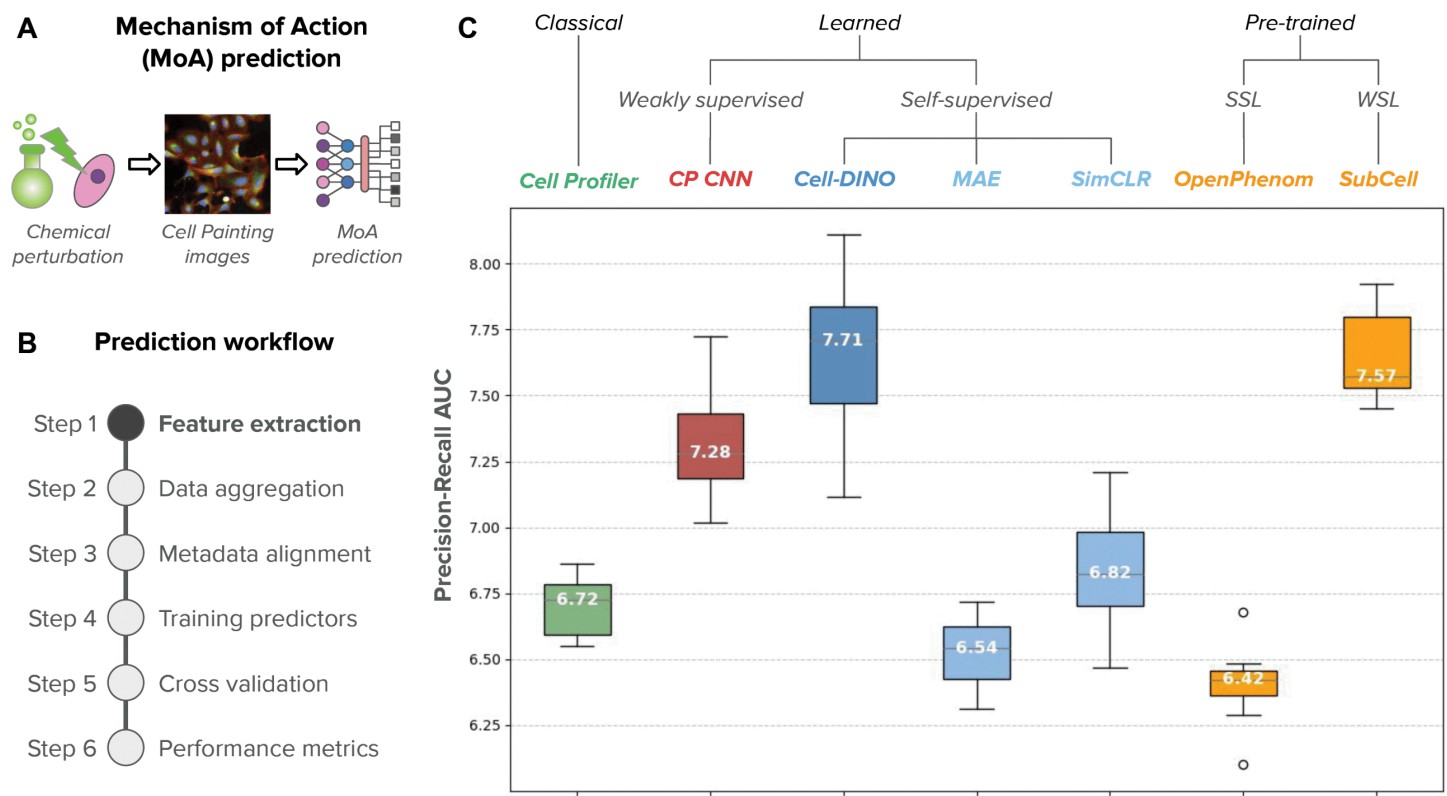

**Fig 7**. **Performance evaluation of Cell-DINO on the CPG datasets.** A) The evaluation task is mechanism of action (MoA) prediction, where chemical compounds are used to treat cells and Cell Painting images are collected to assess their effect and infer their MoA with computational models. B) Illustration of the steps of the computational workflow to predict MoA from Cell Painting images. The only step that changes in these experiments is feature extraction (highlighted in bold). C) Box plots comparing the distributions of performance for the evaluated feature extraction methods. Horizontal axis: evaluated methods organized by feature type. Vertical axis: performance score according to the area under the precision-recall curve of the multi-class MoA prediction problem [62]. Highlighted numbers in the boxes are the median values of the distribution.

challenging task because without explicitly training a parameterized classifier, the feature space needs to be well aligned between training and test samples. Even under these challenges, Cell-DINO embeddings excel and provide better performance compared to other approaches, especially for the protein localization classification problem at the single-cell level (Table 3).

We also find that Cell-DINO embeddings contain variations of cellular morphology that capture spatial and signal-intensity heterogeneity in the HPA-FoV images. We investigated this based on annotations available through the HPA web-portal on spatial and intensity variation of protein localization patterns that reflect population wide consensus patterns [72]. We find that Cell-DINO embeddings are able to cluster single-cells according to their expected heterogeneity (Fig 8). To avoid confounding by cell-line dependent variation, we collected single-cell embeddings for U2OS cells in the HPA-SC dataset, and clustered single-cell features for three example proteins annotated as either presenting no heterogeneity (EFHC2), spatial heterogeneity (PSTPIP2) or intensity heterogeneity (NUSAP) in the protein channel at the single-cell level. Consistently, protein localization appears homogeneous for EFHC2, and heterogeneous for PSTPIP2 and NUSAP according to the similarity matrices of single cells in Fig 8. This demonstrates how Cell-DINO embeddings can detect biologically relevant variation in subpopulations of cells.

In the CPG datasets, we created perturbation-level profiles for 1,571 compounds in the LINCS Cell Painting dataset by aggregating Cell-DINO embeddings. Visual inspection of the cosine similarity matrix suggests these profiles naturally

**Table 3**. **Nearest neighbors search evaluation.** F1-scores obtained with the HPA-FoV and HPA-SC datasets on the protein localization (PL) classification and cell-line (CL) prediction tasks. The SSL Type column reports the self-supervised principle used by the method, and the Supervision column indicates the type of labels used by the model during training.

| Field-of-View (FoV) | | | | |
| --- | --- | --- | --- | --- |
| **Method** | **SSL Type** | **Supervision** | **PL** | **CL** |
| OpenPhenom [38] | Reconstructive | - | 11.5 | 39.3 |
| SimCLR | Contrastive | - | 34.8 | **87.6** |
| SubCell [33] | Reconstructive | Protein ID | 39.0 | 62.3 |
| Cell-DINO | Self-distillation | - | **76.8** | 84.4 |
| Single Cell (SC) | | | | |
| **Method** | **SSL Type** | **Supervision** | **PL** | **CL** |
| OpenPhenom [38] | Reconstructive | - | 17.0 | 50.1 |
| SimCLR | Contrastive | - | 77.4 | 82.8 |
| SubCell [33] | Reconstructive | Protein ID | 78.2 | 79.9 |
| Cell-DINO | Self-distillation | - | **88.5** | **97.4** |

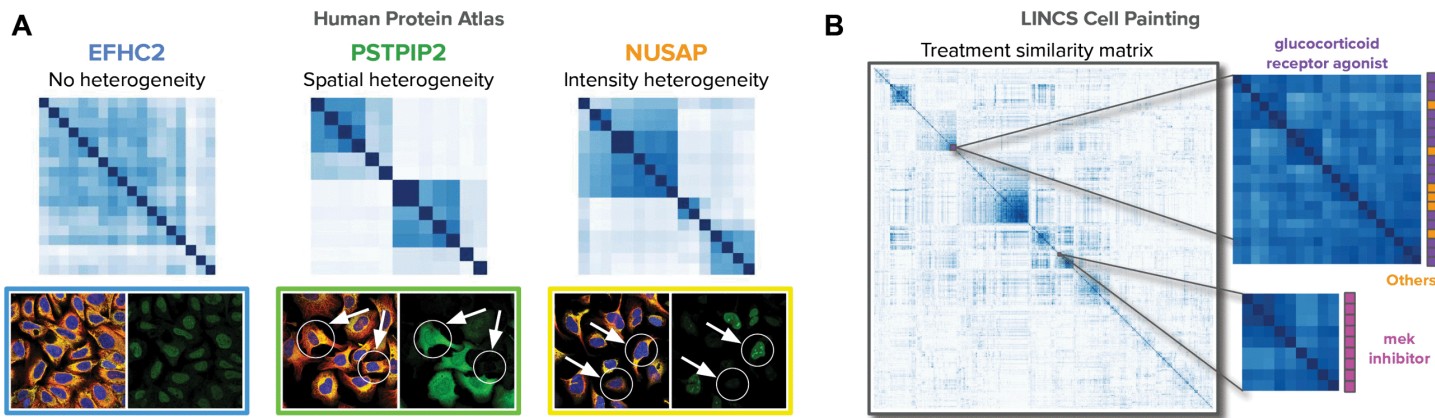

**Fig 8**. **Image-based profiling of cellular morphology.** A) Single-cell heterogeneity analysis on the HPA-SC dataset. Only U2OS cells were selected for this analysis to measure morphological differences for three proteins EFHC2, PSTPIP2, and NUSAP, which have been annotated as exhibiting no heterogeneity, spatial heterogeneity, and intensity heterogeneity, respectively. The matrices display similarity of single cells from images labeled for the corresponding proteins using Cell-DINO embeddings. Color images in the bottom illustrate the composite multi-channel visualization, and green images represent the corresponding protein channel. The arrows point to cells that differ in their protein localization activity, explaining the heterogeneity patterns observed. Larger version available in Fig D in S1 Text) Similarity matrix of treatment-level profiles of aggregated Cell-DINO embeddings for 1,571 compounds in the LINCS dataset. Two compound clusters are highlighted with colored mechanism-of-action annotations on the right-hand side. In all matrices, dark blue indicates high similarity, and light blue or white indicates low similarity.

uncover similarities between perturbations that share similar MoA annotations (Fig 8). The matrix was reordered using hierarchical clustering and displays various sets of compounds grouped together according to their similar morphological effects in cells. Not all the groups found in the matrix are fully annotated with MoA labels and may reveal unknown combinations of MoAs, which gives an opportunity to investigate novel pharmacological relationships. This is in line with quantitative results in Table 1 and Fig 7 that demonstrate Cell-DINO's ability to predict MoAs using existing annotations. Together, these results underpin the inherent biological fidelity of Cell-DINO embeddings, and their potential to enable new and complement existing routes for biological discoveries.

## Cell-DINO finds meaningful local features

A qualitative evaluation of the patch-level features computed by Cell-DINO reveals that they encode meaningful features of cell morphology. We used PCA analysis to reduce the dimensionality of patch-level features for a subset of U2OS

cells in the HPA-SC dataset and visualize the top three principal components (PCs) in the RGB color space (Fig 9A). The resulting visualizations show how different cellular compartments and their morphological properties are encoded in separate PCs. For instance, most of the nucleus variation is encoded in PC-1 (red), while the outer cytoplasm is encoded in PC-2 (green), and the inner cytoplasm is encoded in PC-3 (blue). This local-feature organization explains the ability of Cell-DINO to encode biologically relevant properties in the resulting output embeddings.

In addition to general cellular structure, patch-level features also encode fine-grain, subtle morphological patterns. We evaluated the ability of patch-level features to match more specific morphology across images by computing the cosine similarity between one token in a query image and all tokens in a reference image. The results in Fig 9B show that a token from the nucleus in a query cell with protein localization in the nucleoplasm matches strongly and specifically to tokens in the nucleus of a reference cell with the same protein localization. In a reference cell with no protein localization in the nucleoplasm, the features still match to the nucleus tokens preferentially, but with lower intensity. This indicates that the features encode sufficient information to distinguish general cellular structure as well as their specific and meaningful biological variations.

## Cell-DINO embeddings encode technical variation

We quantitatively evaluate the difference between using Cell-DINO embeddings with and without batch correction on the MoA prediction task with the LINCS Cell Painting dataset. In this experiment, single-cell embeddings are aggregated hierarchically using the mean of features per field of view, then per well, and finally per treatment. Sphering [32,73] is used as the batch correction procedure right after per-well data aggregation, and before treatment-level aggregation. The results indicate that the ability to predict MoAs is reduced by 28% without batch correction (Table 4), resulting in a significant degradation of performance. We used the same procedure to measure the impact of not batch-correcting data in feature embeddings learned by other methods. The results in Table 4 confirm that all image-based embeddings exhibit a similar degradation in performance when batch effects are not addressed (Figs B and C in S1 Text). In this case, batch effects are a negative bias, but in other cases, they can be a positive bias that overestimates predictive performance. Either way, the best practice is to always test for the presence and impact of batch effects to prevent confounded results. Batch effect correction for imaging data has witnessed remarkable progress recently [74,75], but it remains an open problem in the general case, and more research is necessary to design effective and robust batch correction algorithms.

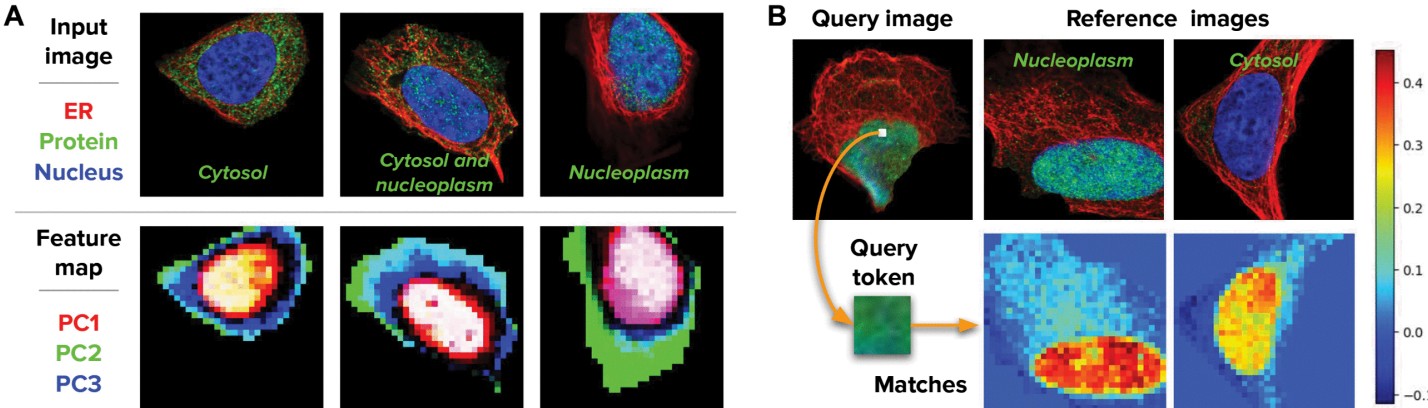

**Fig 9. Patch-level feature maps obtained with Cell-DINO.** A) Example images and feature map visualizations obtained using principal component analysis (PCA) on patch-level features after masking the background. The first three principal components (PCs) are mapped to the RGB color space. B) Matching patch-level features across images using cosine similarity. A query token is selected from the input image and the cosine similarity is calculated against all patches of reference images.

**Table 4**. **Model Performance Comparison With and Without Batch Correction (B.C.) on the LINCS Cell Painting dataset.** The downstream task is MoA classification and the performance metric is PR-AUC. The reported value is the mean of 10 experiments.

| Model | With B.C. | Without B.C. | Percent drop |
|---|---|---|---|
| CP-CNN [32] | 7.32 ± 0.21 | 5.54 ± 0.17 | −24.26% |
| OpenPhenom [38] | 6.40 ± 0.15 | 4.09 ± 0.25 | -36.10% |
| SubCell [33] | 7.65 ± 0.17 | 6.18 ± 0.19 | −19.23% |
| Cell-DINO | 7.67 ± 0.29 | 5.48 ± 0.33 | −28.45% |

## Discussion

Rapid development of methods for automated extraction of visual features is positioning single-cell microscopy as a primary source of phenotypic information. In this work, we present Cell-DINO as an effective self-supervised strategy for learning rich representations of cellular morphology for biological research. In contrast to most contemporary methods, Cell-DINO avoids dependence on prior knowledge and labeled data, and instead yields general-purpose features that can be effectively leveraged for multiple downstream tasks. At the same time, we show that the resulting features are high-quality: on highly competitive benchmark tasks, Cell-DINO achieves similar or better performance when compared to highly tuned end-to-end supervised models, while being substantially more label-efficient. We also find that Cell-DINO offers best-in-class performance compared to other self-supervised learning strategies. Importantly, with state-of-the-art results across competitive benchmark datasets, the task-agnostic nature of Cell-DINO embeddings holds the significant potential to enable the (unsupervised) discovery of unknown biological variation.

By open-sourcing the specialized weights for both HPA and CP Cell-DINO models, our study enables researchers to conduct image-based analysis in pursuit of high-impact biomedical applications, such as protein mislocalization, or large-scale drug screens. Our pre-trained models can be used out-of-the-box, enabling researchers to unlock these avenues even with low-resource computing infrastructure without GPUs. Training Cell-DINO models for new datasets or different image types requires more compute power, but it is a promising way of expanding the impact of imaging in novel directions. In particular, Cell-DINO could be a powerful tool in the context of industrial-scale Cell Painting datasets such as the JUMP-consortium and RxRx-series of datasets [76,77], which we leave as a future research direction. In addition, training models that can adapt to new channel types and image configurations is a frontier research direction that can be powered with the Cell-DINO strategy to produce more generalizable and reusable models [64,78]. This, and the development of user-friendly applications to facilitate the usage of Cell-DINO models by non-power users is also part of our future work.

Beyond the scope of this work is the increasingly appreciated challenge of addressing batch-effects in visual representation learning in biomedicine [79], and in single-cell microscopy in particular [74,75]. We find no evidence that Cell-DINO models are inherently less prone to encoding batch-effects than other approaches. Instead, signs of Cell-DINO models having encoded technical sources of variation are evident e.g. in Fig 3, in Table 4, and in Fig A in S1 Text. Importantly, we show that Cell-DINO embeddings are well-amenable to post-hoc correction methods such as Harmony [52] to remove one or more factors of variation (see Fig 3C). Additionally, we quantitatively evaluated the difference between using Cell-DINO with and without batch correction, confirming that Cell-DINO models capture the inherent technical variation in images, similar to other models and baselines. We leave development of dedicated methods to fortify self-supervised models such as Cell-DINO against confounding by batch-effects to future work.

## Methods

### Datasets

**HPA-FoV dataset.** We analyze subsets of the Human Protein Cell Atlas used for the Kaggle competition "Human Protein Atlas Image Classification". The data is taken from 35 different cell lines with different morphology, such as A549 human

lung adenocarcinoma cells or U2OS human bone osteosarcoma epithelial cells. Each cell culture is treated with fluorescent dyes that bind to the nucleus, microtubules, endoplasmic reticulum, and a protein of interest, providing four image channels with different aspects of cellular morphology. The proteins of interest are divided into 28 classes of protein localizations, annotated by experts. The original goal of the dataset was to test how well protein localization can be predicted given the four channel images.

**HPA-SC dataset.** We also analyzed the HPA Cell Atlas single-cell subset, used for the Kaggle competition "Human Protein Atlas - Single Cell Classification". This dataset is composed of 28 cell lines, and has the same fluorescent channels as in the HPA-FoV classification challenge above, with the protein localizations being divided into 17 locations as well as another "negative" class.

We used the HPA-Cell-Segmentation algorithm [80] recommended in the HPA-SC classification challenge to obtain images of single cells from the FoVs of the main dataset. The algorithm is based on pre-trained U-Nets, and returns both cell segmentation and nucleus segmentation. In the HPA-SC DINO training, we trained the model on full images of single cells regardless of cell size, and inferred the features from center crops of size $512 \times 512$ pixels around the center of each single cell.

**Cell painting.** We also analyzed a collection of Cell Painting datasets, including LINCS [62], JUMP-CP [81] and a combined single-cell resource for training [32]. These datasets were obtained following the Cell Painting protocol [67], where cells are stained with six fluorescent dyes highlighting eight cellular compartments. Images are acquired in five-channels at 20× magnification. Segmentation of Cell Painting images was performed with CellProfiler using a two steps approach: 1) The Hoechst (DNA) channel is segmented with global Otsu thresholding, and used as a prior for the next step. 2) Cell body segmentation with the watershed method in the ER or RNA channel.

The LINCS dataset is a chemical screen of 1,571 FDA approved drugs in six concentrations, tested on A549 cells. In our experiments, we only use samples from DMSO negative controls and the maximum concentration of each compound. The JUMPCP dataset is a processed version of the data made available by the JUMP-Cell Painting Consortium [81]. It contains 229,228 single cell images. We used only the five fluorescence channels in our work. The combined Cell Painting dataset is a collection of subsets from five other Cell Painting datasets, covering gene overexpression perturbations, compound screens, two cell lines (A549 and U2OS), sampled to maximize technical, biological and phenotypic variation. We use this dataset for training feature extraction models.

### Cell-DINO algorithm

At the core, Cell-DINO is the same self-supervised strategy as DINOv2, which is based on the DINO (self-DIstillation with NO labels) algorithm. The adaptations needed to make DINOv2 work with images of cells are primarily intended to accommodate different numbers of channels in the architecture, and to pre-process pixel data according to best practices in microscopy. The self-supervised training algorithm used to create Cell-DINO models does not change with respect to the DINOv2 algorithm [82]. Here, we describe the components of the base DINOv2 algorithm, and then highlight the main adaptations introduced for Cell-DINO.

**DINOv2.** DINO is composed of two networks, student and teacher, that share their architecture. Both networks receive an image and output a feature vector that is then passed on to a projection network that classifies the feature vectors to a logit vector of 65,536 values. The weights of the student network are trained with the cross entropy loss between the outputs of the student and teacher networks. The weights of the teacher network are updated using the exponential moving average of the student network. There are additional losses as part of DINOv2, a Koleo loss and an Ibot loss. After training, the projection head is removed from the feature extractor networks, and the trained teacher network is preserved for feature extraction. The images that the two networks receive are views of the same sample, and augmentations are used to introduce low-level signal alterations that prevent encoding trivial features. Thus, as the two networks are trained to

output a similar feature vector for all views of the same image, the feature extractor learns to be invariant to the changes between the views, and encode only the non-changing aspects of the image.

**Adaptations for Cell-DINO.** Differences in channel number and image content with respect to natural images require to adjust augmentations for biological images. This is done primarily to make sure DINO learns relevant visual features for this type of data. The number of channels of the ViT networks is adjusted independently for each dataset to 4 and 5 channels in the HPA and Cell Painting datasets, respectively. We used cross-validation with a subset of 5,000 FoV images from the HPA dataset to test the effect of introducing different augmentations and trained DINO models for 100 epochs. A linear classification model was then trained on the resulting learned features to evaluate performance. We found the following augmentations to be unnecessary for images of cells: Gaussian blurring, solarization, and greyscale transformations. In addition, we found several other augmentations to be useful: randomly rescaling the intensity of the protein channel (in HPA), dropping the content of a random channel by zeroing out all pixels, replacing color jitter with random brightness and contrast changes to each channel individually. These findings are consistent with a previously reported systematic study of image augmentations for self-supervised learning [83].

Note that augmentations in our work are not meant to address batch effects. In our experiments, we explored the role of basic augmentations as a means to improve performance, but not necessarily remove unwanted technical variation. We did not find evidence that these augmentations make the models robust to batch effects. Prior work has shown that domain-specific augmentations may not be as critical as previously thought for self-supervised learning [84]. In addition, generative models and style transfer techniques have been explored to produce augmented data that explicitly target batch effects [74], which is a more promising approach to mitigate technical variation. We did not explore this approach in our study, and we leave this possibility for future work.

**Computational complexity.** The total training cost of a ViT is associated with three main components: 1) image resolution choices, 2) model size, and 3) training dataset size. Image resolution imposes a cost proportional to image and patch size: the larger the image and the smaller the patch, the more tokens need to be processed by self-attention modules, resulting in complexity that grows quadratically with the number of tokens. Model size multiplies this complexity by the number of layers and dimensionality of hidden representations, and dataset size multiplies complexity again by the number of training images. These three aspects are common to training any ViT, regardless of strategy (supervised vs self-supervised). Cell-DINO follows the same strategy as DINOv2, which increases the complexity of training the ViTs by a factor of approximately two: it simultaneously trains a teacher and a student network (it is less than two in practice because gradients are only computed for the student network). This is the only major source of additional computational cost in the Cell-DINO algorithm compared to other strategies such as supervised ViTs and MAEs. The implementation of DINOv2 has been optimized using the PyTorch framework [85] making it easy to use multi-GPU parallelization, which facilitates scaling when the computational resources are available.

### Vision Transformer (ViT) architecture

The architecture of the student and teacher networks can be any neural network that receives an image and outputs a feature vector. However, DINO shows its strength best with the vision transformer (ViT) [57]. The ViT is based on the transformer architecture [86], which is built as a series of connected attention heads. Each attention head receives a list of $n$ tokens, transforms them into keys, queries and values, and multiplies them in an attention head to get a $n \times n$ attention matrix. While the original transformer was used for natural language processing, where each token is a word, the vision transformer works with images, where each token is the encoding of an image patch of $p \times p$ pixels. One of the benefits of using ViT instead of convolutional neural networks is that the ViT keeps the resolution the same regardless of the depth of the network, allowing us to iteratively calculate more complex features while keeping the same patch-wise resolution.

## Supervised ViTs

The implementation of the supervised ViTs used as baselines for the HPA benchmark follows a standard classification strategy where all samples are expected to label annotations and a loss function minimizes the classification error [57]. Loss function: The training objective minimizes the softmax cross-entropy loss. Optimizer: The supervised ViTs were trained using the AdamW optimizer with parameters $\beta_1 = 0.9$ and $\beta_2 = 0.999$. Learning rate schedule: The learning rate was gradually increased over the first few epochs (warmup) and then linearly decayed for the remainder of training. Data Augmentation: the same augmentations used for self-supervision were employed here.

## Training and evaluation protocol on HPA datasets

We used one ViT model for each dataset. Encoders are ViT-Large networks with default patch size of 16 pixels, or 14 pixels if pre-trained on web images. Transforms used include random contrast (RC) and brightness (RB) augmentations, channel dropping (RCD), flips, random resize crops of size 224. Table 5 shows the impact of these augmentations, especially on protein localization. For decoder (linear layer) training, the following transforms are used : random crop of size 384, flips and self normalization (mean subtraction, division by std). For evaluation, a center crop of size $384 \times 384$ is taken, followed by self normalization. The features used in the linear evaluations consist of the concatenation of a $1 \times 1024$ class token and the average pooling of the output of the last block, resulting into a $1 \times 2048$ feature vector. Unless specified otherwise, we used frozen backbone weights.

## Kaggle evaluation protocol

For the HPA-FoV Kaggle classification challenge, we pre-trained Cell-DINO initialized with DINOv2 weights on full resolution HPA images, using a patch size 14, 518×518 global crops 224×224 local crops for 45K iterations, with batch size of 2048. For fine-tuning, we used a total of 2M samples and optimized the supervised loss with the same configuration.

For evaluation, we used ensembling of five two-layer MLPs. Even with five models, this ensembling uses less parameters than using finetuning. For each val split of the 5-fold dataset, we computed the best threshold per class, similarly to [87]. Then we performed threshold averaging and prediction averaging. Each MLP consists in a 2 layers with attention pooling on the concatenation of the 4 last layers patch tokens. The MLPs were trained on 1036 images, with hidden layer of dimension $4 \times 1024 = 4096$. We detail the performance of different variants in Table 6.

For the HPA-SC Kaggle classification challenge, we used the same the same backbone pre-trained on HPA whole, and one MLP for evaluation without ensembling. The same thresholding strategy was applied.

**Table 5**. **Ablation study on the impact (F1 scores) of augmentations during pretraining on HPA-FoV**. RB: random brightness, RC: random contrast, and RCD: random channel dropping.

| model | Protein loc. | Cell line |
|---|---|---|
| Cell-DINO without RB, RC, RCD | 60.1 | 88.5 |
| Cell-DINO without RCD | 63.7 | 88.7 |
| Cell-DINO | **65.5** | **89.3** |

**Table 6**. **HPA-FoV Kaggle Public scores of protein localization.**

| model | F1 score |
|---|---|
| Cell-DINO +, fine-tuned, linear | 55.0 |
| Cell-DINO + $512 \times 512$, MLP, ensembling | 55.8 |
| Cell-DINO + $512 \times 512$, MLP, best fold | 54.8 |
| Supervised Baseline (CNN [24]) | 54.4 |

## Cell Painting evaluation protocol

As the Cell Painting images are of much lower resolution ($160 \times 160$), we adjusted the parameters accordingly, using ViT small models with patch size 8. We also discarded the local crops in the Cell-DINO strategy as we observed that it was hurting the performance at this resolution. We used random crops of size 128 pixels for training, and center crops of the same size for evaluation. The evaluation procedure was otherwise the same as for HPA on the JUMPCP dataset. The evaluation protocol for Mechanism of Action prediction on LINCS is described in the main text.

## Baselines

We trained several supervised baselines as well as another self-supervised learning strategy, the SimCLR approach. Unless specified otherwise, every baseline uses the same architecture and data augmentation strategies as our approach. We also tried random initialization of the weights or using a DINOv2 model pre-trained on web photographs images. We used the AdamW optimizer, and one cycle learning rate schedule. We performed grid searches varying the learning rate, weight decay and dropout rates for the supervised baselines. We also compare to the DINOv2 approach, where we computed features independently for each channel, replicating the input three times to fit the three-channel input format (RGB) of the open source model [37]. Then we concatenated the features obtained for the four channels and fed this vector as input to a linear layer.

## Computational resources used for training Cell-DINO models

In our experiments, we used standard image resolution of 256x256 pixels for the HPA datasets, and 128x128 pixels for the CPG datasets. Patch size is 16x16 pixels in both cases. We trained ViT-large models, and the sizes of the training datasets are: HPA-FoV 200K images, HPA-SC 500K images, and combined Cell Painting 5M images. For pretraining, Cell-DINO ran for about 12 hours on 32 V100-32G GPUs on the HPA-FoV dataset, and 21 hours for the HPA-SC dataset. On the combined Cell Painting dataset, Cell-DINO ran for 11 hours on 64 V100-32G GPUs. The cost of using a trained Cell-DINO model is the same as any other ViT as there is no additional pre- or post-processing required. We ran feature extraction on 8 GPUs, which typically takes one hour for 0.5M images. We also experimented with training ViT-small models in the exploration phase, which significantly reduces the computational needs by approximately a factor of four. This means that we can train a Cell-DINO ViT-small model with the combined Cell Painting dataset (5M 128x128 images) in approximately 12 hours using 8 GPUs.

## Supporting information

**S1 Text. Supplementary material. Text document with results, figures, tables, and descriptions of additional experiments.**
(PDF)

## Acknowledgments

We thank Alice V. De Lorenci for assistance during our discussions and for experimenting with our models. We also A. Carpenter, S. Singh, M. Babadi, and M. Hirano for helpful discussions. Finally, we thank the DINO team for their code base we build upon in this work, and in particular Marc Szafraniec and Patrick Labatut.

## Author contributions

**Conceptualization:** Armand Joulin, Piotr Bojanowski, Wolfgang M. Pernice, Juan C. Caicedo.

**Data curation:** Michael Doron, Zitong S. Chen, Nikita Moshkov, Wolfgang M. Pernice, Juan C. Caicedo.

**Formal analysis:** Théo Moutakanni, Camille Couprie, Seungeun Yi, Michael Doron, Nikita Moshkov.

**Funding acquisition:** Wolfgang M. Pernice, Juan C. Caicedo.

**Investigation:** Théo Moutakanni, Camille Couprie, Seungeun Yi, Michael Doron, Zitong S. Chen, Nikita Moshkov, Wolfgang M. Pernice, Juan C. Caicedo.

**Methodology:** Théo Moutakanni, Camille Couprie, Michael Doron, Nikita Moshkov, Mathilde Caron, Piotr Bojanowski, Juan C. Caicedo.

**Project administration:** Camille Couprie, Juan C. Caicedo.

**Resources:** Juan C. Caicedo.

**Software:** Théo Moutakanni, Camille Couprie, Seungeun Yi, Michael Doron, Zitong S. Chen, Nikita Moshkov, Elouan Gardes, Mathilde Caron, Hugo Touvron, Wolfgang M. Pernice, Juan C. Caicedo.

**Supervision:** Piotr Bojanowski, Wolfgang M. Pernice, Juan C. Caicedo.

**Validation:** Théo Moutakanni, Camille Couprie, Seungeun Yi, Zitong S. Chen, Nikita Moshkov, Elouan Gardes, Juan C. Caicedo.

**Visualization:** Théo Moutakanni, Camille Couprie, Wolfgang M. Pernice, Juan C. Caicedo.

**Writing – original draft:** Théo Moutakanni, Camille Couprie, Piotr Bojanowski, Wolfgang M. Pernice, Juan C. Caicedo.

**Writing – review & editing:** Théo Moutakanni, Camille Couprie, Piotr Bojanowski, Wolfgang M. Pernice, Juan C. Caicedo.

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
