## [Decision Letter · Decision Letter 0]

8 Apr 2025

PCOMPBIOL-D-25-00185

Cell-DINO: Unbiased Image-based Embeddings for Cell Fluorescent Microscopy

PLOS Computational Biology

Dear Dr. Caicedo,

Thank you for submitting your manuscript to PLOS Computational Biology. After careful consideration, we feel that it has merit but does not fully meet PLOS Computational Biology's publication criteria as it currently stands. Therefore, we invite you to submit a revised version of the manuscript that addresses the points raised during the review process.

Please submit your revised manuscript within 60 days Jun 08 2025 11:59PM. If you will need more time than this to complete your revisions, please reply to this message or contact the journal office at ploscompbiol@plos.org. Please include the following items when submitting your revised manuscript:

We look forward to receiving your revised manuscript.

Kind regards,

Virginie Uhlmann

Academic Editor

PLOS Computational Biology

Mark Alber

Section Editor

PLOS Computational Biology

**Journal Requirements:**

At this stage, the following Authors/Authors require contributions: Theo Moutakanni, Camille Couprie, Seungeun Yi, Michael Doron, Zitong S Chen, Nikita Moshkov, Mathilde Caron, Hugo Touvron, Armand Joulin, Piotr Bojanowsk, Wolfgang M Pernice, and Juan Caicedo. Please ensure that the full contributions of each author are acknowledged in the "Add/Edit/Remove Authors" section of our submission form.

3) Your manuscript is missing the following section: Discussion.  Please ensure all required sections are present and in the correct order. Make sure section heading levels are clearly indicated in the manuscript text, and limit sub-sections to 3 heading levels. An outline of the required sections can be consulted in our submission guidelines here:

5) We have noticed that you have uploaded Supporting Information files, but you have not included a list of legends. Please add a full list of legends for your Supporting Information files after the references list.

Potential Copyright Issues:

i) Figure 6A. Please confirm whether you drew the images / clip-art within the figure panels by hand. If you did not draw the images, please provide (a) a link to the source of the images or icons and their license / terms of use; or (b) written permission from the copyright holder to publish the images or icons under our CC BY 4.0 license. Alternatively, you may replace the images with open source alternatives. See these open source resources you may use to replace images / clip-art:

7) Please amend your detailed Financial Disclosure statement. This is published with the article. It must therefore be completed in full sentences and contain the exact wording you wish to be published.

8) Please ensure that the funders and grant numbers match between the Financial Disclosure field and the Funding Information tab in your submission form. Note that the funders must be provided in the same order in both places as well. Currently, this grant number "2134695 " is missing from the Funding Information tab.

**Reviewers' comments:**

Reviewer's Responses to Questions

Reviewer #1: # Cell-DINO: Unbiased Image-based Embeddings for Cell Fluorescent Microscopy

Dear authors and editors,

I am pleased to review this manuscript, which represents a valuable contribution to the growing intersection of deep learning and biological research—particularly from the authors behind the widely recognised DINO framework. The manuscript is well-structured, clearly written, and demonstrates technical competence. Below, I have structured my feedback according to the review format, beginning with a summary of my overall assessment:

This work introduces Cell-DINO, a self-supervised learning approach based on Vision Transformers, adapted from DINOv2 for the extraction of image-based embeddings from cellular microscopy data. The authors claim that these embeddings are "unbiased" and can be applied across a range of biological tasks without the need for manual annotation. The approach is promising and has the potential to significantly advance bioimage analysis and scalable phenotyping.

Key strengths of the paper include:

- The technical contribution of applying and benchmarking DINOv2 in this context.

- A thorough empirical evaluation.

- Clear writing and a commitment to open science through the public release of code and models.

That said, a few areas require further clarification or improvement:

- The robustness of the embeddings to batch effects remains unconvincing and should be explored in more depth.

- The underperformance of the fine-tuned model (Cell-DINO+) relative to its non-fine-tuned counterpart is unexpected and requires explanation.

- Benchmarking would benefit from broader comparisons, particularly against recent state-of-the-art SSL approaches (e.g., ref 31).

- The paper should provide more details regarding computational resource requirements to improve accessibility for a wider range of researchers.

- Additionally, I noted one minor figure-caption inconsistency and a small wording issue.

Overall, I recommend major revisions to improve the manuscript’s robustness and practical relevance. If batch effects cannot be mitigated, the method’s value for biological applications is limited. Even if they can, the degradation in performance after fine-tuning suggests that users may need to retrain from scratch, creating a barrier for adoption in resource-constrained labs. That said, if these issues—particularly batch effects and fine-tuning—are clearly addressed or thoroughly discussed, the work will still represent a valuable contribution, even without additional experiments.

----

- What are the main claims of the paper and how significant are they for the discipline?

The paper introduces and demonstrates Cell-DINO, a self-supervised learning approach based on Vision Transformers and adapted from the DINOv2 algorithm. It is designed to generate rich, "unbiased" image-based embeddings of cellular morphology from microscopy images, with the goal of enabling scalable and versatile phenotyping without manual annotations. The method is intended to support a wide range of downstream biological analyses, such as protein localization, compound screening, and the recognition and quantification of morphological variations in cells.

These claims have the potential to be highly significant for the field of bioimage analysis, provided that the authors can convincingly demonstrate that these "unbiased embeddings" are indeed robust to confounding factors such as batch effects, which could otherwise limit the method’s validity. While the authors apply Harmony to account for biological variation (specifically cell-line-dependent variation), this post hoc correction does not fully address broader technical batch effects (e.g., plate-to-plate or lab-to-lab variability). I would encourage the authors to address this aspect explicitly. For instance, prior studies such as ref 29 acknowledge batch effects, ref 36 incorporates mitigation strategies, and ref 46 attempts to analyze technical biases. In large-scale efforts such as the Human Protein Atlas (HPA), where this method could be highly valuable, ensuring robustness to batch effects is essential.

The manuscript mentions that "augmentations are used to indicate which information is not important," but it does not appear that these augmentations were designed specifically to mitigate batch variation. Additionally, manually selecting augmentations may introduce human bias at a different level, which slightly contrasts with the paper’s emphasis on minimizing bias stemming from human preconceptions. For instance, Figure 2C shows multiple clusters of mitochondria and others, potentially hinting at unexplored batch or technical variation.

Furthermore, the broader impact of this work will depend on its accessibility to the biological research community. Its significance hinges on whether pre-trained models can be directly applied by other laboratories or whether retraining on new datasets is feasible with widely available computing resources. At present, there is no mention of the computational resources required. Based on the model complexity, it appears that substantial infrastructure (e.g., approximately 8 A100 GPUs running for two weeks per model) and expertise would be necessary. I recommend that the authors explicitly report training requirements and suggest ways to lower the barrier for adoption.

- Are these claims novel? If not, which published articles weaken the claims of originality of this one?

The application of DINOv2 to biomedical imaging data, specifically in cell microscopy, alongside benchmarking, constitutes a technically meaningful and relatively new contribution. I appreciate the authors’ effort and investment in adapting a state-of-the-art algorithm like DINOv2 for bioimage analysis and for providing a thorough empirical evaluation.

That said, the broader claim—that self-supervised learning (SSL) outperforms classical feature engineering and supervised methods for microscopy image analysis—is well-supported in prior literature and is cited. Furthermore, the use of transformer-based SSL in bioimage analysis is not entirely novel, as similar approaches have been explored (e.g., with DINO, SimCLR, and MAE), although DINOv2 itself has not been applied in this domain.

In summary, while the technical contribution of applying and benchmarking DINOv2 is valuable, the conceptual advancement is relatively incremental when compared to existing SSL work in bioimage analysis. If the authors could demonstrate that these embeddings can be directly clustered into biologically meaningful groups without further correction, the novelty would be more pronounced. As it stands, the term "unbiased" seems to primarily refer to the absence of manual labels, rather than addressing technical confounding factors like batch effects.

- Are the claims properly placed in the context of the previous literature? Have the authors treated the literature fairly?

The paper references relevant prior work, but the comparative discussion could be expanded. For example, there is no direct comparison between Cell-DINO and the implementation in ref 29, which would be highly relevant. Ref 31 benchmarks against ref 29 as well as against an earlier version of this submission. Additionally, ref 46 trains DINO, MAE, and SimCLR on Cell Painting data, but the authors do not compare Cell-DINO to those results.

It is also noteworthy that ref 31 made concerted efforts to outperform the earlier Cell-DINO submission. It would strengthen the manuscript if the authors discussed how Cell-DINO compares not only to their own baselines (including an MAE trained with augmentations optimized for Cell-DINO) but also to these newer, closely related methods, including the MAE from ref 31.

- Do the data and analyses fully support the claims? If not, what other evidence is required?

The current data and analyses support several claims regarding model performance. However, there is no direct evidence that the embeddings are robust to batch effects or other technical biases. As mentioned earlier, prior works (e.g., ref 29, ref 36, ref 46) provide examples of how to analyze and mitigate such issues.

In addition to clarifying the terminology (e.g., "unbiased"), I would encourage the authors to explore whether augmentations or alternative strategies could reduce batch effects. Such an investigation could represent a meaningful contribution to the field. If no such strategies are identified, this may suggest that DINO, in its current form, may not be fully optimal for tasks requiring invariance to batch variation.

- Would additional work improve the paper? How much better would the paper be if this work were performed and how difficult would it be to do this work?

Yes, additional work would enhance the paper, both in terms of technical rigor and its practical utility for the broader research community.

1. Computational cost and accessibility (low effort): The authors should explicitly report the computational resources required to train, fine-tune, and deploy Cell-DINO. It would also be helpful to clarify whether these demands are comparable to DINOv2 and to discuss whether the method is primarily suited to well-resourced laboratories or can be more broadly adopted through pre-trained models. Additionally, in Figure 4, while comparing against a single CNN from a top ensemble is reasonable, a direct comparison with the full ensemble in terms of computational cost would provide a fuller picture of performance vs. resource trade-offs.

2. Clarification of fine-tuning effects (low effort): The manuscript should explain why the fine-tuned Cell-DINO+ model underperforms relative to the non-finetuned version (as shown in Figure 3). This raises important questions about whether fine-tuning may, in some cases, degrade learned representations, and whether users would be better served by using the released model as-is or by retraining from scratch.

3. Benchmarking and baseline selection (moderate effort): The benchmarking could be strengthened by including direct comparisons to more recent or competitive methods, such as those presented in ref 31, which outperforms an earlier version of Cell-DINO. Additionally, clarifying the rationale for selecting MAE and SimCLR as baselines would be helpful. Addressing the omission of other SSL methods, such as MoCo, BYOL, or SimSiam, would provide reassurance that the chosen baselines are representative of the broader SSL landscape.

4. Batch effect analysis (substantial effort, uncertain success): A deeper evaluation of whether Cell-DINO embeddings encode batch effects would significantly improve the manuscript’s biological relevance. Although Harmony is applied to address biological variability, further work is needed to assess and mitigate technical confounders such as batch, plate, or site effects, which are common in large-scale microscopy datasets.

5. Augmentation strategy and impact on robustness (moderate-to-high effort, promising impact): Given the importance of augmentations in SSL frameworks like DINO and SimCLR, the authors should justify their choice of using the same augmentation pipeline across all methods and explain how the selected augmentations balance biological variability and technical bias. Investigating whether augmentations (or learnable augmentation strategies) could mitigate batch effects would be a valuable addition.

- PLOS Computational Biology encourages authors to publish detailed protocols and algorithms as supporting information online. Do any particular methods used in the manuscript warrant such treatment? If a protocol is already provided, for example for a randomized controlled trial, are there any important deviations from it? If so, have the authors explained adequately why the deviations occurred?

The training and evaluation protocols are well-documented, and the code will be made available. However, the manuscript would benefit from including detailed documentation on resource requirements and practical guidelines for reproducibility, particularly for laboratories without access to extensive computational infrastructure.

- Are original data deposited in appropriate repositories and accession/version numbers provided for genes, proteins, mutants, diseases, etc.?

N/A.

- Has the author-generated code that underpins the findings been made publicly available?

The authors state that the code will be made available on GitHub upon publication.

- Does the study conform to any relevant guidelines such as CONSORT, MIAME, QUORUM, STROBE, and the Fort Lauderdale agreement?

N/A.

- Are details of the methodology sufficient to allow the experiments to be reproduced?

Yes.

- Is any software created by the authors freely available?

Yes.

- Is the manuscript well organized and written clearly enough to be accessible to non-specialists?

Overall, yes. The manuscript is well-structured and clearly written. However:

1. The term "unbiased embeddings" could be potentially misleading for non-specialists and would benefit from clarification.

2. The word "adaptation" may not be entirely precise here, as the modifications to DINOv2 appear relatively minor, and this does not fully align with the concept of domain adaptation. The term "application" might more accurately describe the use of DINOv2 in this context. In interdisciplinary research, ensuring clarity in technical terminology is particularly important to avoid potential misunderstandings among readers from diverse backgrounds.

Minor issues:

1. In Figure 4, the caption states "horizontal lines," but the lines are vertical.

2. In "Data Availability", the phrase "as the Cell-DINO model weights…" should likely be "as well as the Cell-DINO model weights."

- Does the paper use standardized scientific nomenclature and abbreviations? If not, are these explained at the first usage?

Yes.

Reviewer #2: In their manuscript Moutakani et al. investigate the potential application and advantages of using the DINOv2 model to study cellular morphology. Cellular morphology in bioimages remain a challenging topic and, despite important improvements since the integration of DL in analyses pipeline in recent years, can still greatly benefit from new and advanced methods to improve further. This is attested by the amount of recently published literature and the regular organization of competitions. It also feels important to note that progress made by research teams dedicated to image analysis is sometimes having a hard time reaching biologist research teams due to both usage complexity and difficulty of retraining the models for their own images.

The Cell-DINO method presented here provides an interesting solution to bypass the difficulty of producing ground truth datasets. This has the potential of simplifying the application of such a powerful DL method to end users in biology labs.

The manuscript compares the Cell-DINO efficiency to 2 other supervised and self-supervised when applied to 2 publicly available image datasets. These results are showing better results, especially in the low training data range which again has the potential of simplifying the uptake of this method by end users.

Overall, this manuscript demonstrates that CellDINO is a useful and promising tool for studying biological questions in the context of imaging and the authors are sharing the code in a FAIR format to reproduce their results. However, making CellDINO more readily applicable to other datasets and more easy to reuse for less computer savvy labs could help make it more impactful.

Finally, I would consider important to see more models compared, in order to confirm that CellDINO is consistently performing better than supervised and self-supervised approaches.

Main comments:

- The code is conveniently shared in a git repository that will be open after publication. It also describes expected run times for a specific hardware for the cell-dino-hpa. This part is clearer and more developed than the cell-painting-eval that would benefit from reaching the same quality level (with direct links to the notebooks, and expected runtimes and hardware). Expected runtimes are especially interesting since the authors recommends using the CPU version of the TF package to prevent issues with GPU drivers because TF 2.12 is an old version.

- The code is well presented and organized but do not possess a jupyter notebook implementation or a graphical user interface such as a napari plugin or the one provided by the Cellpose authors. In it's current shape, the reach of this study could remain limited to only labs with highly skilled computer scientists. Would it be possible to try and offer a more graphical approach to limit this technical step as much a possible and maximize the reach of this algorithm?

- Adding more models for comparison and explaining better why their were chosen would reinforce the paper's results

# Authors summary and introduction

- the authors categorize the "current approaches to analyze cell microscopy images as relying on classical features carefully chosen [...] or supervised strategies". I believe this may have to be updated with the recent [SAM in bioimage analysis paper](https://www.nature.com/articles/s41592-024-02580-4) as well as the [Transformers do not outperform Cellpose](https://www.biorxiv.org/content/10.1101/2024.04.06.587952v1) paper.

# Cell-DINO encodes diverse biological properties of cells

- In Fig 2B: The subtitle "single cells - training datasets" is unclear. I guess the authors are not talking about the dataset used for training, so what is it meant to describe? In general the figure 2 legend could be more clear. The figure 2D is not used in the main text.

- The claim that "protein localization patterns are largely invariant across human cell lines" should be accompanied by a citation.

- The note that "harmonized HPA-FoV embeddings qualitatively resemble the structures identified by supervised methods" is a very interesting claim, but the authors should make this comparison simpler for the reader by putting both figures side-by-side or, if possible, quantitatively assess it.

# Cell-DINO outperforms supervised ViT results

- The chosen supervised model doesn't seem to be described here. Please describe it in details and explain this choice. In case it is a single model, considering how many such models exist, I'm not sure this comparison is sufficient to support the claim that Cell-DINO generally outperforms supervised ViT.

- Considering that Cell-DINO+ outperforms Cell-DINO in both pre-trained and fine-tuned conditions for protein-localization, could the authors discuss why they favor Cell-DINO with random initialization?

- The figure lacks a way to compare Cell-DINO results fine-tuned with Cell lines or Protein loc. to Cell-DINO+ to fully be able to understand the effect of the model initialization

# Cell-DINO is competitive against highly tuned models

- The BestFitting and DualHead solutions described in the text are not highlighted in the figure. Instead "Kaggle" is displayed as "a single supervised CNN". Please make these solutions clearly appear.

# Cell-DINO outperforms alternative self-supervised strategies

- Considering that Cell-DINO heavily outperforms MAE but not so much simCLR, it would be interesting to have more models to compare here.

- When claiming that Cell-DINO improve upon the performance of state-of-the-art approaches, please cite the results directly in the manuscript

# Cell-DINO reduces the dependency on manual annotations

- In the HPA SC context: the conclusions of this paragraph are confusing, although I agree that the Fig.5 supports the claim of better results in the low training data range, it's unclear where the 24% and 72% relative improvements are calculated from. It seems to me that the results are fetched from both the protein and cell line graphs and always picking the most striking difference only, in that case it may seem unfair not to take into account the other less striking differences.

- An additional word should be said about the comparison between Supervised and CellDINO in the HPA SC - Protein context where both model F1 scores seem to meet at the 100% sample usage.

- Finally it's unclear why the HPA SC dataset is more challenging. Considering the results in Fig3 results, I would have expected the Protein localisation studies to be the most challenging independently from the FoV/SC status. Could the authors explain further their assessment?

# Cell-DINO enables predictions in unbiased imaging experiments

# Cell-DINO excels at unbiased profiling of cellular morphology

- Table 2: Cell-DINO+ is used in this table again, I'm not sure to understand why it is only used here and not in the other comparisons?

- Table 2: If Cell-DINO+ is excluded from this table then Cell-DINO is indeed showing the best results in 3 out 4 contexts, however it seems to only be compared to a single other model. The text is stating "other approaches", would it be possible to add more models here?

- Fig7.: the images shown are not sufficient to provide a good example of the expected spatial or intensity heterogeneity. The images are too small to conclude and the arrows are not clearly pointing to specific cells or part of cells. Please consider adding a supplementary figure for these cells and an explanation of the expected results.

# Conclusion

I fully agree with the authors that Cell-DINO is an effective strategy with the great benefits of creating less biased embeddings and requiring less annotated data. However, I'm not sure the general conclusion that Cell-DINO "outperforms supervised approaches and other self-supervised strategies" is supported enough by their comparisons considering that only a couple of these approaches/strategies were compared and that displayed results are only showing a better efficiency in some cases compared to the other models (see Fig 3-4-5, Table 1-2), a more nuanced phrasing may be more adapted or adding additional models could also further strengthen their conclusion.

# Methods

- Cell Painting: Would it be interesting to test for lower drug concentration with expected milder effects? Considering the good performance of Cell-DINO in creating unbiased and accurate embeddings, I would be curious to see if it outperforms the other methods when differences are less pronounced.

- In the methods: When describing Cell-DINO algorithm, it's unclear how the DINOv2 algorithm differs from Cell-DINO.

Minor notes:

- An effort should be made to harmonize the data set namings: CL/Cell Lines, Protein localization/Prot loc./Protein/PL/Protein loc./protein-localization, HPA-SC/HPA single cell, etc...

- Figure 3 is very complex to read and should be reorganized to support the author's hypothesis better

- In general, figures are complex to read and should be improved.

Reviewer #3: Points 1–5 are typological in nature; points 6–8 are more critical in scope:

1. Clarify Metric in Figure 3 Caption

It would be helpful to explicitly state in the Figure 3 caption that the metric being plotted is the F1-score, to avoid ambiguity for readers unfamiliar with the evaluation protocol.

2. Legend Label in Figure 4

In Figure 4, the label “DINO: 0.55” (blue) could be renamed to “Cell-DINO: 0.55” for consistency with terminology used throughout the paper and to distinguish it clearly from DINOv2 or other models. Also, there is a minor typo in “Kaggle: 0:54” — this should be corrected to “Kaggle: 0.54”.

3. Clarify Acronyms in Table 1

In the Table 1 caption, the acronym AUPRC could be spelled out as “Area Under the Precision-Recall Curve”, especially since it is used only for the Cell Painting tasks listed at the bottom of the table.

4. Repeat Augmentation Definitions in Table 3 Caption

Although RB (random brightness), RC (random contrast), and RCD (random channel dropout) are defined earlier in the text, repeating these definitions in the Table 3 caption would improve clarity and make the table more self-contained for standalone reading.

5. Add a Dedicated Section on Methodological Differences with DINOv2

The paper would benefit from a clearer, dedicated comparison between Cell-DINO and DINOv2. As a reader, it was initially unclear what aspects of the approach are novel or domain-specific (e.g., input channels, augmentations, data modality) versus what is inherited from DINOv2.

The section “Cell-DINO Algorithm” helps but could be split into two subsections—one briefly describing the base DINOv2 method, and one clearly listing the adaptations introduced in Cell-DINO. An explicit breakdown would help highlight the novelty of the contributions.

6. Clarify and Justify the Term “Unbiased Embeddings”

The term “unbiased embeddings” is used extensively but is not formally defined or empirically quantified. Specifically:

There is no explicit test of invariance to known sources of bias in microscopy, such as:

-Batch effects

-Illumination differences

-Image transformations (e.g., flips, intensity rescaling)

The claim of "unbiasedness" is currently inferred from downstream performance and qualitative visualizations (e.g., UMAPs).

A more formal framing or even a simple empirical test of robustness would strengthen this claim and improve interpretability.

7. Deeper Exploration of the Augmentation Space

While Table 3 offers useful insight into specific augmentations, a more exhaustive exploration of the augmentation space would add significant value. For example, similar to Figure 5 in the SimCLR paper, a systematic study could help identify which augmentations are most beneficial or detrimental for microscopy, and whether they generalize across datasets or phenotyping tasks.

8. Discussion of Failure Modes of CellDINO + Interpretability of Embeddings

It would be helpful to include a brief discussion on:

Failure modes of Cell-DINO: Are there particular classes (e.g., rare protein localizations or cell lines) where performance degrades?

Interpretability of the learned embeddings: Do they encode organelle shape? Texture? Intensity distribution?

Even simple saliency maps or patch-level attention heatmaps could offer insight into what the model is focusing on, which is especially important in biomedical settings.

**Have the authors made all data and (if applicable) computational code underlying the findings in their manuscript fully available?**

Reviewer #1: Yes

Reviewer #2: Yes

Reviewer #3: Yes

PLOS authors have the option to publish the peer review history of their article (what does this mean?). If published, this will include your full peer review and any attached files.

Reviewer #1: No

Reviewer #2: **Yes: **Sébastien Herbert

Reviewer #3: **Yes: **Manan Lalit

**Figure resubmission:**
---

## [Decision Letter · Decision Letter 1]

30 Sep 2025

PCOMPBIOL-D-25-00185R1

Cell-DINO: Self-Supervised Image-based Embeddings for Cell Fluorescent Microscopy

PLOS Computational Biology

Dear Dr. Caicedo,

Thank you for submitting your manuscript to PLOS Computational Biology. After careful consideration, we feel that it has merit but does not fully meet PLOS Computational Biology's publication criteria as it currently stands. Therefore, we invite you to submit a revised version of the manuscript that addresses the points raised during the review process.

Please submit your revised manuscript within 30 days Nov 30 2025 11:59PM. If you will need more time than this to complete your revisions, please reply to this message or contact the journal office at ploscompbiol@plos.org. Please include the following items when submitting your revised manuscript:

We look forward to receiving your revised manuscript.

Kind regards,

Virginie Uhlmann

Academic Editor

PLOS Computational Biology

Mark Alber

Section Editor

PLOS Computational Biology

**Journal Requirements:**

1) Please ensure that the funders and grant numbers match between the Financial Disclosure field and the Funding Information tab in your submission form. Note that the funders must be provided in the same order in both places as well. Currently, the order of this grant number ( 2348683 ) does not match in both the places.

2) The file inventory includes multiple files for Figure 4. We would recommend either combining these into a single Figure 4.tiff file with separate internal panels, or renumbering them as individual figures, as we are not able to publish multiple components of a single figure as separate files.

**Reviewers' comments:**

Reviewer's Responses to Questions

Reviewer #1: The authors have addressed all of the concerns I raised, and I believe the manuscript can be published in its current form. I appreciate their thoughtful responses and commend their commitment to open science. This work will be a valuable contribution to the field of bioimage analysis.

Reviewer #2: I thank the authors for carefully considering my comments in their revised manuscript. All of my major concerns have been addressed, and I now have only a few minor observations.

Minor observations:

- In the cell-painting-eval/README file: "feature extraction need to be performed with the DINOv2 implementation provided here (link to be added soon)". I expect this code will be released and linked as soon as the paper is accepted to make the pipeline fully reproducible.

- A variable seems to be misnamed in the file located at cell-painting-eval/LINCS/run_all.sh. Line 12 should read `echo "Error: $CONFIG does not exist."` instead of `echo "Error: $File does not exist."` as to correctly point to the problematic file.

- I tried to follow the conda environment creation protocol and ended with a pip dependency conflicts both on a Windows and an Ubuntu machine. I would encourage the authors to share a yml recipe to streamline the environment creation as they did in the cell-dino-hpa repo.

- In the cell-dino-hpa code repo, the main title is still showing the old article title ("# Cell-DINO: Unbiased Image-based Embeddings for Cell Fluorescent Microscopy")

Reviewer #3: 1. Naming:

I recommend considering renaming Cell-DINO to Cell-DINOv2. This would align better with future iterations (e.g., Cell-DINOv3 if adapted to DINOv3). Alternatively, if the differences between Cell-DINO and DINOv2 are minimal, the work could be presented as an investigation of how well DINOv2 applies to fluorescent microscopy images. This would help with clarity and long-term consistency. (This would of require re-working Figure 3, but it might make it clear to the reader that what you are presenting is not a new method, but rather the same method DinoV2 but investigated with different initializations and pre-trained on microscopy data and not natural images).

2. Abstract:

The abstract could be made clearer. Currently, it opens with:

"Accurately quantifying cellular morphology at scale..."

This is more of a downstream application. The core contribution is that the method learns representations that can then enable such quantification/classification. Re-framing the abstract to emphasize “representation learning” upfront would sharpen the message for readers.

3. Title:

While the representations are indeed learned in a self-supervised manner, the evaluation still requires training a linear classifier on a small set of labeled data. To reflect this more precisely, consider a title such as:

“Self-supervised representation learning for fluorescent microscopy with minimal supervised fine-tuning.”

This would set accurate expectations for the reader.

4. Figure 1:

(i) I suggest splitting the figure into two individual figures: (A,B) illustrating the method, and (C) showing the datasets. The dataset figure can stand independently from the method.

(ii) In the ViT schematic (B), the sequence of tokens should remain consistent (16 in input, 16 in output), and the attention matrix should reflect 16 × 16 to avoid confusion.

5. Figure 2:

It would be very helpful to show clustering of single cells for protein localization in the HPA dataset before applying Harmony, for readers to fully appreciate the effect of the correction.

6. Figure 3: You say in the caption "Protein localization blue shades and Cell line green shades", but in the legend on the bottom right it looks the other way around (blue for cell line, green for protein localization). Please clarify.

7. Grammar and Style Corrections:

(i) Author Summary: remove the space in “Cell-DINO ,” → “Cell-DINO,”

(ii) Introduction: >"because the goal is to uncover the effect of such perturbations."  reference missing and replaced by question mark currently.

(iii) Figure 3: “Worse” → “Worst”.

(iv) Figure 3 caption: “All reported results are the F1 score” → “F1 scores”.

(v) Cell-DINO outperforms alternative self-supervised strategies: "Mask Autoencoder" should be "Masked Autoencoder".

(vi) Table 3: "Self-distilation" should be "Self-distillation".

(vii) Figure 4: "BestFitting" should be "bestfitting" in the legend.

(viii) Cell-DINO excels at image-based profiling of cellular morphology: >"We find that Cell-DINOembeddings ..." -> add a space after Cell-DINO

8. Overall:

I appreciate that the revised version de-emphasizes the claim of “unbiasedness” of the embeddings, which makes the framing more balanced.

The writing can be worked upon more. Currently I feel as a reader that the writing can be tightened in several places.

**Have the authors made all data and (if applicable) computational code underlying the findings in their manuscript fully available?**

Reviewer #1: Yes

Reviewer #2: **No: **Training and feature extraction with the DINOv2 implementation doesn't seem to be provided yet

Reviewer #3: Yes

PLOS authors have the option to publish the peer review history of their article (what does this mean?). If published, this will include your full peer review and any attached files.

Reviewer #1: **Yes: **Qin Yu

Reviewer #2: **Yes: **Sébastien Herbert

Reviewer #3: **Yes: **MANAN LALIT

**Figure resubmission:**
---

## [Editor Report · Decision Letter 2]

9 Dec 2025

Dear Dr. Caicedo,

We are pleased to inform you that your manuscript 'Cell-DINO: Self-Supervised Image-based Embeddings for Cell Fluorescent Microscopy' has been provisionally accepted for publication in PLOS Computational Biology.

Best regards,

Virginie Uhlmann

Academic Editor

PLOS Computational Biology

Mark Alber

Section Editor

PLOS Computational Biology

---

## [Editor Report · Acceptance letter]

PCOMPBIOL-D-25-00185R2

Cell-DINO: Self-Supervised Image-based Embeddings for Cell Fluorescent Microscopy

Dear Dr Caicedo,

I am pleased to inform you that your manuscript has been formally accepted for publication in PLOS Computational Biology. Your manuscript is now with our production department and you will be notified of the publication date in due course.

With kind regards,

Judit Kozma
